# DUAL: Learning Diverse Kernels for Aggregated Two-sample and Independence Testing

**Zhijian Zhou**[*§]  **Xunye Tian**[*§]  **Liuhua Peng**[§]  **Chao Lei**[§]
**Antonin Schrab**[†]    **Danica J. Sutherland**[‡]    **Feng Liu**[§]
University of Melbourne[§]    University of Cambridge[†]    UBC & Amii[‡]
{zhijianzhou.ml, xunyetian.ml}@gmail.com
liuhua.peng@unimelb.edu.au
clei1@student.unimelb.edu.au
afls2@cam.ac.uk    dsuth@cs.ubc.ca    fengliu.ml@gmail.com

## Abstract

To adapt kernel two-sample and independence testing to complex structured data, aggregation of multiple kernels is frequently employed to boost testing power compared to single-kernel tests. However, we observe a phenomenon that directly maximizing multiple kernel-based statistics may result in highly similar kernels that capture highly overlapping information, limiting the effectiveness of aggregation. To address this, we propose an aggregated statistic that explicitly incorporates *kernel diversity* based on the covariance between different kernels. Moreover, we identify a fundamental challenge: a *trade-off* between the diversity among kernels and the test power of individual kernels, i.e., the selected kernels should be both effective and diverse. This motivates a testing framework with selection inference, which leverages information from the training phase to select kernels with strong individual performance from the learned diverse kernel pool. We provide rigorous theoretical statements and proofs to show the consistency on the test power and control of Type-I error, along with asymptotic analysis of the proposed statistics. Lastly, we conducted extensive empirical experiments demonstrating the superior performance of our proposed approach across various benchmarks for both two-sample and independence testing.[¶]

## 1 Introduction

In modern machine learning, non-parametric hypothesis tests have become essential for comparing probability distributions and detecting statistical dependencies without imposing restrictive model assumptions. Kernel-based methods provide a powerful framework for these tasks by embedding probability distributions into reproducing kernel Hilbert spaces (RKHS), enabling rigorous yet flexible measures of discrepancy and dependence [1, 2]. For example, the Maximum Mean Discrepancy (MMD) is a prominent kernel two-sample test metric used to determine whether two sets of observations originate from the same distribution [1, 3–12]. Similarly, the Hilbert–Schmidt Independence Criterion (HSIC) is a related method designed to measure statistical dependence between random variables, thus serving as a test of independence [2, 13–16]. They use kernel methods to enhance statistical power and are widely adopted in machine learning fields including domain adaptation [17, 18], generative modeling [19], adversarial learning [20], machine-generated text detection [21, 22], causal discovery [23], semi-supervised representation learning [24], continual learning [25], and more.

**Related works.** Kernel aggregation, combining multiple kernels into a single test procedure, has proven to be highly effective in non-parametric hypothesis testing, often yielding substantial gains

---

[*]Equal contribution. [¶]Code: https://github.com/tmlr-group/MMD-HSIC-DUAL.

39th Conference on Neural Information Processing Systems (NeurIPS 2025).

in statistical power [7, 9, 26]. This is because the choice of single kernel is critical: even though a well-chosen kernel can greatly enhance a test's ability to detect departures from the null hypothesis, a poorly chosen kernel may fail to capture the relevant differences. To mitigate the risk of selecting a suboptimal kernel, a common strategy is to aggregate test statistics across a collection of kernels rather than relying on any single kernel. Such multi-kernel aggregation approaches have been widely adopted in various hypothesis testing scenarios – including two-sample testing [7–9, 27, 28] and independence testing [7, 27, 29, 30] – and have consistently demonstrated improved test power (i.e., the probability of correctly rejecting the null hypothesis under the alternative) and adaptivity. In these works, aggregating kernels with different bandwidths or characteristics enables the tests to capture a broad range of potential data structures, often achieving higher power than single-kernel methods.

**Motivations.** Aggregating multiple statistics[2] with diverse kernels is a powerful approach to capture complex distributional characteristics and achieve higher test power in hypothesis testing. However, our research uncovers a core limitation that, contrary to conventional wisdom, not all kernels contribute to the test power, and simply using more kernels does not always imply higher power. In other words, the inclusion of uninformative or redundant kernels can potentially reduce the effectiveness of the test. As illustrated in Figure 1, we evaluate the test power of the two-sample test using a set of 20 kernels, implemented as [28], and also perform the test using randomly selected subsets of 5 kernels. Our results demonstrates that the performance varies significantly across different subsets, and in some cases, the aggregation over a small subset of kernels can indeed outperform the aggregation of full set of kernels, indicating that indiscriminately ensembling various kernels may introduce redundancy that dilutes the test's power. Thus, our work highlights the importance of identifying and utilizing informative kernels rather than relying on a large, potentially redundant collection.

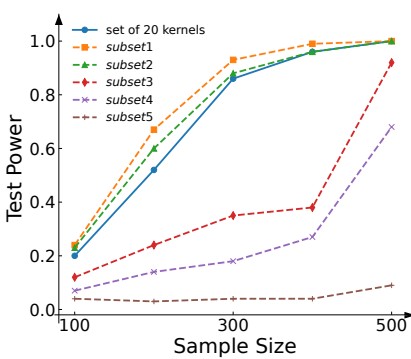

**Figure 1:** Comparing the test power of aggregating different sets of kernels in the two-sample testing problem on the BLOB dataset. The solid blue line shows the performance when aggregating all 20 kernels. The five dotted lines represent the test power when aggregating five different randomly selected subsets (each containing 5 kernels).

**Contributions.** We propose a new kernel aggregating methods in hypothesis testing, *Diverse U-statistic Aggregation with Learned kernels (DUAL)*, which improves kernel-based nonparametric tests (e.g., MMD and HSIC) by introducing a notion of *diversity* inspired by ensemble learning, where diversity among base models is crucial for performance [33–35]. In particular, DUAL computes the covariance matrix of $U$-statistics obtained from multiple kernels and leverages it to quantify the pairwise diversity among these kernels. Building on this diversity measure, we develop a novel test statistic that integrates each kernel's $U$-statistic with the pairwise diversity between kernels, thereby capturing complementary information and improving test sensitivity. Furthermore, we employ post-selection inference in the testing procedure to adaptively maximize test power through informed kernel selection while rigorously controlling the Type-I error rate. As a result, DUAL provides a general power enhancement for both two-sample and independence tests—instantiated as MMD-DUAL and HSIC-DUAL—which harnesses the benefits of aggregating multiple kernels while mitigating the influence of uninformative or weak kernels. We provide theoretical guarantees for the proposed approach (including valid Type-I error control and improved asymptotic power) and present extensive empirical validation on diverse benchmarks, demonstrating that DUAL-based tests achieve strong performance relative to existing state-of-the-art methods.

## 2 Preliminaries

In this section, we provide background about the non-parametric hypothesis testing problems that we are interested in, including both *MMD two-sample testing* and *HSIC independence testing*. To begin, we introduce the concept of the second-order $U$-statistic with a kernel $\kappa$, which is a key statistical tool. Suppose we have a random sample $W = \{w_1, w_2, \ldots, w_n\}$ from some distribution $\mathbb{W}$ on a separable metric space $\mathcal{W}$. Let $h(w_1, w_2; \kappa)$ be a symmetric function of two arguments defined over

---

[2]In this work, we *aggregate multiple test statistics into a single statistic* [9, 28, 31], initially via a sum for simplicity, eventually via a weighted $\ell_2$ norm. This differs multiple testing across different kernels [8, 29, 32].

the kernel $\kappa$. The second-order $U$-statistic, with computational complexity $O(n^2)$, is defined as

$$U_n^\kappa(W) = \binom{n}{2}^{-1} \sum_{1 \le i_1 < i_2 \le n} h(\boldsymbol{w}_{i_1}, \boldsymbol{w}_{i_2}; \kappa) . \tag{1}$$

For two-sample and independence tests, the widely used kernel-based test statistics MMD and HSIC, respectively, can each be formulated within the framework of $U$-statistics, which we introduce below. See [36] for a detailed introduction to these, and [31] for an overview of MMD/HSIC testing results.

**Two-sample Test with MMD.** Let $\mathbb{P}$ and $\mathbb{Q}$ denote two unkown Borel probability measures over an instance space $\mathcal{X} \subseteq \mathbb{R}^d$, and draw samples $X = \{\boldsymbol{x}_i\}_{i=1}^n \sim \mathbb{P}^n$ and $Y = \{\boldsymbol{y}_j\}_{j=1}^n \sim \mathbb{Q}^n$. We aim to test the null hypothesis $H_0 : \mathbb{P} = \mathbb{Q}$ against the alternative hypothesis $H_1 : \mathbb{P} \ne \mathbb{Q}$. The key step in this test is to quantify the discrepancy between distribution $\mathbb{P}$ and $\mathbb{Q}$. Writing $W = \{\boldsymbol{w}_i\}_{i=1}^n = \{(\boldsymbol{x}_i, \boldsymbol{y}_i)\}_{i=1}^n$, an unbiased estimate of the squared MMD [3] with kernel $\kappa$ is

$$\mathrm{MMD}_{n,\kappa}^2(W) = \binom{n}{2}^{-1} \sum_{1 \le i_1 < i_2 \le n} h_{\mathrm{MMD}}^{(\kappa)}(\boldsymbol{w}_{i_1}, \boldsymbol{w}_{i_2})$$

$$h_{\mathrm{MMD}}^{(\kappa)}((\boldsymbol{x}, \boldsymbol{y}), (\boldsymbol{x}', \boldsymbol{y}')) = \kappa(\boldsymbol{x}, \boldsymbol{x}') + \kappa(\boldsymbol{y}, \boldsymbol{y}') - \kappa(\boldsymbol{x}, \boldsymbol{y}') - \kappa(\boldsymbol{y}, \boldsymbol{x}').$$

**Independence Test with HSIC.** Let $\mathbb{U}_{xy}$ be a Borel probability measure defined on the space $\mathcal{X} \times \mathcal{Y}$, and draw an i.i.d. sample $(X, Y) = \{(\boldsymbol{x}_1, \boldsymbol{y}_1), (\boldsymbol{x}_2, \boldsymbol{y}_2), \dots, (\boldsymbol{x}_m, \boldsymbol{y}_m)\} \sim \mathbb{U}_{xy}^m$. Denote by $\mathbb{U}_x$ and $\mathbb{U}_y$ the marginal distributions of $\boldsymbol{x}$ and $\boldsymbol{y}$ respectively. We aim to test the null hypothesis $H_0 : \mathbb{U}_{xy} = \mathbb{U}_x \times \mathbb{U}_y$ (independence) against the alternative hypothesis $\mathbb{U}_{xy} \ne \mathbb{U}_x \times \mathbb{U}_y$ (dependence). Let $n = \lfloor m/2 \rfloor$ and $W = \{\boldsymbol{w}_i\}_{i=1}^n = \{(\boldsymbol{x}_i, \boldsymbol{x}_{i+n}, \boldsymbol{y}_i, \boldsymbol{y}_{i+n})\}_{i=1}^n$. To measure the discrepancy between $\mathbb{U}_{xy}$ and $\mathbb{U}_x \mathbb{U}_y$, we estimate the HSIC with kernels $\gamma$ on $\mathcal{X}$ and $\ell$ on $\mathcal{Y}$ as[3]

$$\mathrm{HSIC}_n^{(\gamma,\ell)}(W) = \binom{n}{2}^{-1} \sum_{1 \le i_1 < i_2 \le n} h_{\mathrm{HSIC}}^{(\gamma,\ell)}(\boldsymbol{w}_{i_1}, \boldsymbol{w}_{i_2})$$

$$h_{\mathrm{HSIC}}^{(\gamma,\ell)}((\boldsymbol{x}_1, \boldsymbol{x}_2, \boldsymbol{y}_1, \boldsymbol{y}_2), (\boldsymbol{x}_1', \boldsymbol{x}_2', \boldsymbol{y}_1', \boldsymbol{y}_2')) = \tfrac{1}{4} h_{\mathrm{MMD}}^{(\gamma)}((\boldsymbol{x}_1, \boldsymbol{x}_2), (\boldsymbol{x}_1', \boldsymbol{x}_2')) \, h_{\mathrm{MMD}}^{(\ell)}((\boldsymbol{y}_1, \boldsymbol{y}_2), (\boldsymbol{y}_1', \boldsymbol{y}_2')) .$$

For consistency with MMD, we will refer to the product kernel $\kappa((\boldsymbol{x}, \boldsymbol{y}), (\boldsymbol{x}', \boldsymbol{y}')) = \gamma(\boldsymbol{x}, \boldsymbol{x}')\ell(\boldsymbol{y}, \boldsymbol{y}')$ as "the kernel" of HSIC, as justified by HSIC's relationship to MMD [3, Thm. 25].

**Degeneracy.** MMD and HSIC are both used to assess the difference between two distributions (i.e., $\mathbb{P}, \mathbb{Q}$ and $\mathbb{U}_{xy}, \mathbb{U}_x \times \mathbb{U}_y$) [1–3]. Correspondingly, in the context of two-sample and independence tests, the null hypotheses take the form $H_0$: *the two distributions are identical*, while the alternative hypotheses are formulated as $H_1$: *the two distributions differ*. The $U$-statistics underlying MMD and HSIC exhibit a common structural property: they are *first-order degenerate* under the null hypothesis $H_0$, and typically *non-degenerate* under the alternative hypothesis $H_1$. Formally, a $U$-statistic with function $h(\cdot; \kappa)$ is said to be *first-order degenerate* if its conditional expectation satisfies

$$h_1(\boldsymbol{w}_1; \kappa) = \mathbb{E}[h(\boldsymbol{w}_1, \boldsymbol{w}_2; \kappa) \mid \boldsymbol{w}_1] = 0.$$

If $\mathrm{Var}_{\boldsymbol{w}_1}(h_1(\boldsymbol{w}_1; \kappa)) \ne 0$, the $U$-statistic is classified as *non-degenerate*. This degeneracy structure plays a critical role in determining the asymptotic distribution of the test statistics under the null and alternative hypotheses, as well as in the design of our proposed test statistic.

## 3 Motivation

In this section, we will identify a phenomenon consistently observed across prior aggregation methods: while individually strong kernels are necessary, aggregating redundant kernels can degrade performance, and including weak kernels adds little useful information.

**Aggregating better kernels may not give higher performance.** Intuitively, one might expect that an aggregation of powerful kernels would outperform each kernel on its own. However, the empirical results tell a more nuanced story. For example, in Figure 2a we observe that, for three high-performing

---

[3]We use the second-order HSIC estimate on $\lfloor m/2 \rfloor$ quadruples [7] for its greater convenience over the complete unbiased HSIC fourth-order $U$-statistic (which is also computable in quadratic time) [37].

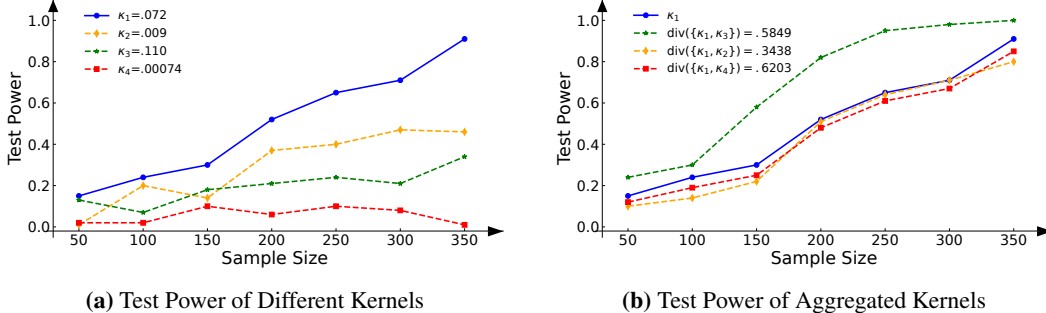

**(a)** Test Power of Different Kernels

**(b)** Test Power of Aggregated Kernels

**Figure 2:** Test power versus samples size on BLOB dataset. (a) The performance of four different individual kernels with different bandwidths. (b) The performance of aggregating the first kernel $\kappa_1$ with each of the kernels. The diversity[4] between $\kappa_1$ and $\kappa_4$ is the largest, and that between $\kappa_1$ and $\kappa_2$ is the smallest.

kernels $\kappa_1, \kappa_2$ and $\kappa_3$, kernel $\kappa_2$ achieves higher test power than kernel $\kappa_3$. Now, consider probably the simplest aggregation method: combine the features used by two kernels, corresponding to using the kernel which is their sum. In this case, we have simply that $U_n^{\{\kappa_1,\kappa_2\}} = U_n^{\kappa_1} + U_n^{\kappa_2}$. Even though $\kappa_2$ is stronger than $\kappa_3$, the combination $\{\kappa_1, \kappa_3\}$ attains greater test power than $\{\kappa_1, \kappa_2\}$ (Figure 2b).

The reason for this seemingly counter-intuitive result lies in the diversity of information that different kernels contribute. In the context of $U$-statistic-based tests, each kernel $\kappa$ produces a test statistic $U_n^\kappa$ that reflects a particular view of the data. If two kernels are highly correlated in the information they capture, their $U$-statistics will also be highly correlated. In our example, the test statistics $U_n^{\kappa_1}$ and $U_n^{\kappa_2}$ are more strongly correlated with each other than $U_n^{\kappa_1}$ and $U_n^{\kappa_3}$. Consequently, $\kappa_1$ and $\kappa_2$ exhibit lower diversity: *they redundantly capture similar aspects of the underlying distributional difference*. Aggregating redundant kernels (as in $\{\kappa_1, \kappa_2\}$) offers little new information beyond what $\kappa_1$ already provides. By contrast, $\kappa_1$ and $\kappa_3$ are less correlated, so $\kappa_3$ contributes complementary information that $\kappa_1$ alone misses, making the combined statistic $U_n^{\{\kappa_1,\kappa_3\}}$ more informative.

**Quality of kernels matters as well.** However, *diversity alone is not sufficient to guarantee high performance*. The kernels must also be effective in detecting the effect of interest. To illustrate, consider kernel $\kappa_4$, which is highly diverse relative to $\kappa_1$. Among the pairs we consider, $\{\kappa_1, \kappa_4\}$ has the greatest diversity in the sense of capturing very different statistical features. Nonetheless, as seen in Figure 2b, the aggregated $\{\kappa_1, \kappa_4\}$ does not perform as well as $\{\kappa_1, \kappa_3\}$, nor $\{\kappa_1, \kappa_2\}$. The reason is that $\kappa_4$ is a 'weak' kernel: its individual test power is too low across sample sizes (Figure 2a), which indicates that it cannot provide enough useful information to detect the pattern differences. This highlights that an ineffective kernel, no matter how different, can drag down the performance of an otherwise strong aggregation. Thus, high test power from an aggregated test statistic arises when the component kernels are *both individually powerful and mutually complementary*.

## 4 Aggregating $U$-Statistic with Diversity

As introduced in the Section 3, it is crucial to introduce diversity into the multiple kernel aggregation. Here, we propose the *multivariate U-statistic*.

**Multivariate $U$-Statistic.** Given a constant integer $c > 1$, let $\mathcal{K} = \{\kappa_1, \kappa_2, ..., \kappa_c\}$ denote a set of $c$ different kernels. To incorporate information from multiple kernels, we construct a multivariate $U$-statistic by aggregating the individual $U$-statistics defined in (1) with each kernel in $\mathcal{K}$. For a random sample $W$, the resulting multivariate $U$-statistic is computed with complexity $\mathcal{O}(cn^2)$ as:

$$\boldsymbol{U}_n^{\mathcal{K}}(W) = \left( U_n^{\kappa_1}(W), U_n^{\kappa_2}(W), ..., U_n^{\kappa_c}(W) \right)^T. \tag{2}$$

Given the potential redundancy in the information captured by different kernels, we investigate the diversity among multiple kernels and integrate their contributions in a manner that accounts for

---

[4]Motivated by [38, 39], the relative diversity value for kernel $\kappa_i$ with $i \in \{2, 3, 4\}$ w.r.t. $\kappa_1$ is computed as $\left( 1 + |\mathrm{Cor}(U_n^{\kappa_1}(W), U_n^{\kappa_i}(W))| \sqrt{\mathrm{Var}(U_n^{\kappa_1}(W))/\mathrm{Var}(U_n^{\kappa_i}(W))} \right)^{-1}$. Here, the term $\mathrm{Cor}(\cdot, \cdot)$ denotes the Pearson correlation and the square root term serves as a scaling factor when comparing the relative diversity of kernels $\kappa_2, \kappa_3$, and $\kappa_4$ with respect to kernel $\kappa_1$.

their mutual dependencies. Specifically, the diversity between $\kappa_a$ and $\kappa_b$ for $1 \leq a, b \leq c$ can be characterized by the (co)variance $\sigma_{a,b}$ of $U_n^{\kappa_a}(W)$ and $U_n^{\kappa_b}(W)$. In practice, the true (co)variance is unknown and must be estimated from the observed sample $W$, which poses a particular challenge for MMD and HSIC. Under the null hypothesis $H_0$, these $U$-statistics are first-order degenerate, where the second-order (co)variance dominates and scales as $O(n^{-2})$. In contrast, under the alternative hypothesis $H_1$, the $U$-statistics are non-degenerate, and the first-order (co)variance dominates, scaling as $O(n^{-1})$. Consequently, the naive rescaling of the (co)variance by $n$ yields convergence to 0 under $H_0$, whereas rescaling by $n^2$ results in convergence to $+\infty$ under $H_1$. This discrepancy highlights the difficulty of constructing a (co)variance estimator that consistently converges to a finite, non-zero limit across both hypotheses, which is essential in enabling meaningful comparison and aggregation of kernel-based test statistics under both null and alternative hypotheses.

Fortunately, we can always apply the second-order (co)variance estimator on samples from the null hypothesis to investigate the diversity among multiple kernels, motivated by [28].[5] For MMD and HSIC, samples under the null hypothesis can be simulated by resampling the observed data $W$ with replacement. Building on this, we utilize the simulated null samples $W_{H_0} = \{w_i'\}_{i=1}^n$ to assess the diversity between $\kappa_a$ and $\kappa_b$ by computing the second-order (co)variance estimator [40–42] between $n \cdot U_n^{\kappa_a}(W_{H_0})$ and $n \cdot U_n^{\kappa_b}(W_{H_0})$ with computational complexity $O(n^2)$ as

$$n^2 \cdot \hat{\sigma}_{H_0,a,b} = n^2 \binom{n}{2}^{-2} \sum_{1 \leq i_1 < i_2 \leq n} h(w_{i_1}', w_{i_2}'; \kappa_a) h(w_{i_1}', w_{i_2}'; \kappa_b) . \tag{3}$$

Given the covariance matrix $\widehat{\Sigma}_{H_0}$ with entries $n^2 \cdot \hat{\sigma}_{H_0,a,b}$, we integrate the contributions of multiple $U$-statistics (i.e., $\boldsymbol{U}_n^{\mathcal{K}}(W)$) in a manner that accounts for mutual dependencies among kernels as

$$T_n^{\mathcal{K}}(W) = n^2 \left(\boldsymbol{U}_n^{\mathcal{K}}(W)\right)^T \widehat{\Sigma}_{H_0}^{-1} \boldsymbol{U}_n^{\mathcal{K}}(W) \tag{4}$$

which is inspired by the *Mahalanobis distance* [43], as done in [28]. In this work, we assume that the covariance matrix $\widehat{\Sigma}_{H_0}$ is strictly positive-definite. Notably, the dimension of the covariance matrix is decided by the number of kernels, i.e., $c$, with a computational complexity $\mathcal{O}(c^3)$, independent of the sample size $n$; thus computational complexity remains low as $n$ increases. The asymptotic properties of the multivariate $U$-statistic (2), the aggregated statistic (4), and the estimated covariance matrix $\widehat{\Sigma}_{H_0}$ are provided in Appendix A.1.

**Remark 1.** *In our multivariate $U$-statistic (4), each dimension of $n\boldsymbol{U}_n^{\mathcal{K}}(W)$ is normalized to a common scale using matrix of covariances of the different kernels. This normalization is crucial to ensure that kernels with varying scales or magnitudes do not disproportionately influence the statistic, thereby mitigating potential biases of aggregation.*

## 5 Two-sample and Independence Testing with Learned Diverse Kernels

In this section, we introduce the implementation pipeline of DUAL, which follows the *data-splitting approach* for kernel selection [6, 44–46]. Even though data-splitting will reduce the sample size in the testing procedure, learning diverse and powerful kernels can *gain extra power* and *adapt to various datasets*. We partition the dataset into a training set, $W_{tr} = \{w_{tr,i}\}_{i=1}^n$, and a testing set, $W_{te} = \{w_{te,i}\}_{i=1}^n$. For notational convenience, we assume both sets contain $n$ elements.[6]

### 5.1 Learning Multiple Diverse Kernels

The selection of kernels that maximize the aggregated statistic $T_n^{\mathcal{K}}(W)$ effectively minimizes the (co)variances of the $U$-statistics under the null hypothesis while maximizing the $U$-statistics under the alternative hypothesis. This enhances the power of hypothesis testing, as discussed in Section 1. Thus, given training samples $W_{tr} = \{w_{tr,i}\}_{i=1}^n$, the kernel set $\mathcal{K}$ is learned as

$$\{\kappa_1, \kappa_2, ..., \kappa_c\} \in \arg\max\{T_n^{\mathcal{K}}(W_{tr})\} , \tag{5}$$

where $T_n^{\mathcal{K}}(W_{tr})$ is defined as in (4) but computed based on the training samples $W_{tr}$. We apply a gradient method [48, 49], initialized with a set of distinct kernels, to maximize the aggregated

---

[5]The original method in [28] focuses specifically on MMD with a tailored (co)variance estimator, while we generalize the approach to a broader class of $U$-statistics by employing a unified (co)variance estimator.

[6]One can aggregate over multiple splits, as in [47], but this may not actually help compared to a single split.

statistic over pre-specified kernels, following previous approaches [50, 51].[7] The learned kernels are independent of testing samples, ensuring that their use in testing does not violate the type-I error constraint.

**Remark 2.** *When $c = 1$, the statistic (4) becomes the square of the signal-to-noise ratio of that single kernel. This was the objective proposed in previous work to select a single MMD [5, 6] or HSIC [15, 16] kernel; thus (5) reduces to those methods when $c = 1$.*

## 5.2 Testing with both Diverse and Powerful Kernels

After learning kernels to optimize the diversity within our multivariate $U$-statistics, we would further like to select only the effective kernels. We focus on kernels that effectively capture evidence against the null hypothesis (under which test statistics have mean zero), while placing less emphasis on kernels that demonstrate limited performance in subsequent testing.

**Extracting prior knowledge.** In order to identify and select effective kernels within our optimized aggregation, we extract essential sign information from the training data—computed as the signum vector of the decomposed statistic $T_n^{\mathcal{K}}(W_{tr})$, inspired by [46]. Specifically, we decompose $\widehat{\Sigma}_{H_0} = \widehat{\boldsymbol{L}}_{H_0}\widehat{\boldsymbol{L}}_{H_0}$ using the Schur method[8] [52] with computational complexity $O(c^3)$, yielding:

$$T_n^{\mathcal{K}}(W_{tr}) = n^2(\boldsymbol{U}_n^{\mathcal{K}}(W_{tr}))^T\widehat{\Sigma}_{H_0}^{-1}\boldsymbol{U}_n^{\mathcal{K}}(W_{tr}) = n^2 \left\|\widehat{\boldsymbol{L}}_{H_0}^{-1}\boldsymbol{U}_n^{\mathcal{K}}(W_{tr})\right\|_2^2, \tag{6}$$

where $\|\cdot\|_2$ denotes the $\ell_2$ norm. $\widehat{\boldsymbol{L}}_{H_0}^{-1}$ *helps reduce correlations among different kernels* [46]. Then, to select strong kernels that capture complementary information and contribute to the final aggregated statistic, we first investigate the information from training samples by computing the *signum vector* as

$$\boldsymbol{F}_{tr} = \text{sgn}\left(\widehat{\boldsymbol{L}}_{H_0}^{-1}\boldsymbol{U}_n^{\mathcal{K}}(W_{tr})\right) \in \{-1, +1\}^c,$$

where $\text{sgn}(\cdot)$ is signum function as $\text{sgn}(\boldsymbol{a}) = (\text{sgn}(a_1), \text{sgn}(a_2), \cdots, \text{sgn}(a_d))^T$ for a vector $\boldsymbol{a} = (a_1, a_2, \cdots, a_d)^T$, and $\text{sgn}(a_i) = a_i/|a_i|$ for $a_i \neq 0$; otherwise, $\text{sgn}(a_i) = 1$.

**Selection inference based on extracted knowledge.** Based on the learned $\mathcal{K}$, we perform the test using the testing samples $W_{te} = \{\boldsymbol{w}_{te,i}\}_{i=1}^n$.[9] The corresponding aggregated statistic is defined as

$$T_n^{\mathcal{K}}(W_{te}) = n^2(\boldsymbol{U}_n^{\mathcal{K}}(W_{te}))^T\widehat{\Sigma}_{H_0}^{-1}\boldsymbol{U}_n^{\mathcal{K}}(W_{te}) = n^2 \left\|\widehat{\boldsymbol{L}}_{H_0}^{-1}\boldsymbol{U}_n^{\mathcal{K}}(W_{te})\right\|_2^2, \tag{7}$$

which is defined analogously to Eqn. (4), but computed based on the testing samples $W_{te}$.

In a similar manner, we infer the signum vector of testing samples, i.e., $\boldsymbol{F}_{te} = \text{sgn}\left(\widehat{\boldsymbol{L}}_{H_0}^{-1}\boldsymbol{U}_n^{\mathcal{K}}(W_{te})\right)$. Given the two signum vectors $\boldsymbol{F}_{tr}$ and $\boldsymbol{F}_{te}$, we can calculate an indicator vector $\boldsymbol{F}$ to assess the alignment between the training and testing signum vectors as follows

$$\boldsymbol{F} = \{F_i\}_{i=1}^c \in \{0, +1\}^c \quad \text{with} \quad F_i = \mathbb{I}[F_{te,i} = F_{tr,i}], \tag{8}$$

where $F_{te,i}$ and $F_{tr,i}$ are $i$-th elements of $\boldsymbol{F}_{te}$ and $\boldsymbol{F}_{tr}$, respectively. The alignment vector $\boldsymbol{F}$ selects the components of the aggregated statistic that share the same signum value across the training and testing samples. Given the selection, we focus on the components that are more likely to capture deviations from the null hypothesis, and define the test statistic with selection inference as

$$\mathcal{T} = n^2 \left\|\boldsymbol{F} \odot \widehat{\boldsymbol{L}}_{H_0}^{-1}\boldsymbol{U}_n^{\mathcal{K}}(W_{te})\right\|_2^2, \tag{9}$$

where $\odot$ is the element-wise product.

---

[7]Prior work on choosing single MMD or HSIC kernels [5, 6, 15, 16] has emphasized the importance of maximizing the power of a test, rather than maximizing the statistic. Our statistic, however, is already studentized, which makes directly estimating the test power less practical (Theorem 5) but also removes the incentive to e.g. simply scale $\kappa$ to $C\kappa$ for some large $C$; our statistic is invariant to such changes. Also see Remark 2.

[8]The Schur decomposition works for positive-definite matrices, and in this paper, we assume the covariance matrix $\hat{\Sigma}_{H_0}$ is positive-definite. In practice, to ensure positive definiteness, we can replace $\hat{\Sigma}_{H_0}$ with $\hat{\Sigma}_{H_0} + \lambda \boldsymbol{I}$, where $\lambda > 0$ is a small regularization constant and $\boldsymbol{I}$ denotes the identity matrix.

[9]Notably, for statistic (6), we compute $\widehat{\Sigma}_{H_0}$ based on $W^{H_0}tr$, which is resampled from $W^{H_0}tr$. The entries of $\widehat{\Sigma}_{H_0}$ are calculated as in Eqn.(3) with $W_{tr}^{H_0}$. Similarly, for statistic (7), $\widehat{\Sigma}_{H_0}$ is computed based on $W_{te}^{H_0}$, which is resampled from $W_{te}$. For notational simplicity, we omit the explicit dependence on $W_{tr}^{H_0}$ and $W_{te}^{H_0}$.

**Remark 3.** *Our approach differs from previous bi-directional hypothesis testing [46], which constructs rejection regions along the directions of $\boldsymbol{F}$ and $-\boldsymbol{F}$ to determine if the test statistic lies within these regions. In contrast to their method, which employs an additional parameter calibrated via a separate validation dataset to adjust the significance levels of the rejection regions, our technique avoids such parameter tuning. Additionally, our method diverges from post-selection inference approaches [53, 54], which select significant statistics based solely on predefined kernels without incorporating insights from the training phase. Such post-selection inference methods are constrained to specific kernel classes and utilize less precise "streaming" estimators, potentially leading to lower-powered tests when using fixed kernels [55, 56].*

**Wild Bootstrap for Testing Threshold.** To obtain the testing threshold $\tau_\alpha$ for a significance level $\alpha$, we employ the wild bootstrap for $\mathcal{T}$ with Rademacher random variables [57, 58, 7, 59]. Specifically, let $B$ be the iteration number of bootstraps. In the $b$-th iteration ($b \in [B]$), we generate i.i.d. Rademacher variables $\boldsymbol{\epsilon} = (\epsilon_1, \ldots, \epsilon_n)$, that is, $\Pr(\epsilon_i = 1) = \Pr(\epsilon_i = -1) = 1/2$ for $i \in [n]$, and then compute

$$\boldsymbol{U}_n^{\mathcal{K},b}(W_{te}) = \binom{n}{2}^{-1} \sum_{1 \leq i_1 < i_2 \leq n} \epsilon_{i_1} \epsilon_{i_2} \boldsymbol{h}(\boldsymbol{w}_{i_1}, \boldsymbol{w}_{i_2}; \mathcal{K}) \,,$$

where $\boldsymbol{h}(\boldsymbol{w}_1, \boldsymbol{w}_2; \mathcal{K}) = (h(\boldsymbol{w}_1, \boldsymbol{w}_2; \kappa_1), ..., h(\boldsymbol{w}_1, \boldsymbol{w}_2; \kappa_c))^T$.

Correspondingly, we calculate the $b$-th alignment vector $\boldsymbol{F}^b$, analogous to Eqn. (8), as follows

$$\boldsymbol{F}^b = \{F_i^b\}_{i=1}^c \in \{0, +1\}^c \quad \text{with} \quad F_i^b = \mathbb{I}[F_{te,i}^b = F_{tr,i}] \text{ and } \boldsymbol{F}_{te}^b = \text{sgn}\big(\widehat{\boldsymbol{L}}_{H_0}^{-1} \boldsymbol{U}_n^{\mathcal{K},b}(W_{te})\big) \,,$$

and we perform selection inference with $\boldsymbol{F}^b$ to derive the $b$-th wild bootstrap statistic as

$$\mathcal{T}^b = n^2 \left\| \boldsymbol{F}^b \odot \widehat{\boldsymbol{L}}_{H_0}^{-1} \boldsymbol{U}_n^{\mathcal{K},b}(W_{te}) \right\|_2^2 \,.$$

Taking the original test statistic in Eqn. (9) as $\mathcal{T}^{B+1}$, we estimate the testing threshold (i.e., $(1-\alpha)$-quantile of the null distribution of the test statistic with selection inference) as follows

$$\hat{\tau}_\alpha = \inf \left\{ \tau \in \mathbb{R} : 1 - \alpha \leq \frac{1}{B+1} \sum_{b=1}^{B+1} \mathbb{I}[\mathcal{T}^b \leq \tau] \right\} \,. \tag{10}$$

Finally, we propose the test with the testing threshold $\hat{\tau}_\alpha$ and the test statistic $\mathcal{T}$ in Eqn. (9) as

$$\mathfrak{h}(X, Y; \kappa) = \mathbb{I}[\mathcal{T} > \hat{\tau}_\alpha] \,, \tag{11}$$

where $\mathfrak{h}(X, Y; \kappa) = 1$ means the null hypothesis is rejected; otherwise, it is accepted.

The computational complexity of the above testing procedure is $\mathcal{O}(Bcn^2 + n^2c^2 + c^3)$. The subsequent theorem characterizes the behavior of the test statistic under both the null and alternative hypotheses.

**Theorem 1.** *Let $\mathcal{K}$ be a collection of bounded characteristic kernels. Under the null hypothesis $H_0$, the test in (11) has type-I error bounded by $\alpha$, i.e., $\Pr_{H_0}(\mathfrak{h}(X, Y; \kappa) = 1) \leq \alpha$, even non-asymptotically. Meanwhile, under any fixed alternative hypothesis $H_1$, and assuming Assumption 1 (in Appendix B.1) holds, the test has power converging to 1, i.e., $\lim_{n \to \infty} \Pr_{H_1}(\mathfrak{h}(X, Y; \kappa) = 1) = 1$.*

In Theorem 1, we validate the test by proving that the type-I error is controlled at level $\alpha$ in a non-asymptotic sense under $H_0$, and the test power approaches one asymptotically under fixed $H_1$. The asymptotic properties of the test statistic with selection inference are provided in Appendix A.2.

## 6 Experiments

### 6.1 Datasets & Baselines

We evaluate our proposed methods on benchmarks for two-sample and independence testing. For two-sample testing, we use three datasets: a frequently used synthetic BLOB dataset [45, 51, 5, 6], the MNIST (versus generative adversarial model DCGAN [60]) dataset [6–8], and the ImageNet (versus ImageNetV2 [61]) dataset [6, 46]. For independence testing, we consider the Higgs dataset (a high-dimensional physics dataset) [62], MNIST, and CIFAR-10. These benchmarks encompass a diverse range of data modalities and difficulties, providing a rigorous evaluation of our tests.

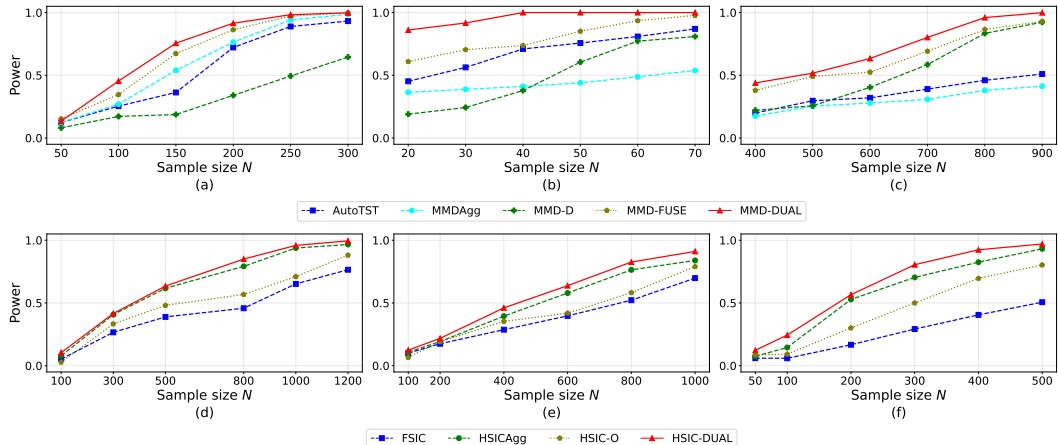

**Figure 3:** Two-sample $(a-c)$ experiments on dataset BLOB, MNIST and ImageNet; and independence $(e-f)$ experiments on dataset Higgs, MNIST and CIFAR10. The power results are averaged over 1,000 repetitions and the type-I error are all controlled under the significant level $\alpha = 0.05$, where the type-I error experiments can be found in the supplementary material.

We compare MMD-DUAL (for two-sample tests) and HSIC-DUAL (for independence tests) against both standard baselines and recent state-of-the-art methods. In the two-sample case, baseline methods include the AutoML two-sample test (AutoTST) [63], the MMD Aggregated test (MMD-Agg) [8], the MMD two-sample test with deep kernel (MMD-D) [6], and the recently proposed multi-kernel test MMD-FUSE [9]. For independence testing, baselines include the Finite Set Independence Criterion (FSIC) [13], an aggregated HSIC test (HSIC-Agg) [7], and independence testing with optimized bandwidth (HSIC-O). All methods are calibrated to control the Type-I error at $\alpha = 0.05$ for a fair comparison. As shown in Figure 3, the proposed MMD-DUAL and HSIC-DUAL consistently outperform all baselines across the six benchmarks. In every case, our adaptive methods achieve the highest test power (rejection rate under the alternative), demonstrating a clear advantage over both classical and contemporary methods on both two-sample (BLOB, MNIST, ImageNet) and independence (Higgs, MNIST, CIFAR-10) tasks. The detailed description of datasets, baselines, experimental settings can be found in the Appendix C.

## 6.2 Ablation Study

**Effectiveness of Diverse Kernels and Selection Inference**. To quantify the effect of learning diverse kernels and selection inference in our proposed DUAL, we conduct a series of well-designed ablation study on all the benchmarks. We only display the results of BLOB for MMD-DUAL and Higgs for HSIC-DUAL, and the results on other datasets can be found in Appendix C. Figure 4 (a) and (e) report the test power (for level $\alpha = 0.05$) for the full DUAL methods and three ablated variants of each: (i) AU+D: without selection—a variant without the selection inference in the testing procedure (i.e., all candidate are always aggregated, but still optimizing the diversity between kernels in the training procedure), (ii) AU+S: without introducing diversity—a variant without the diverse kernel pool (using an fixed identity covariance matrix) while still performing the selection inference technique, and (iii) AU: plain aggregation—a baseline that uses multiple kernels in an aggregated two-sample and independence testing with neither diversity nor selection inference enhancements. We observe that both MMD-DUAL and HSIC-DUAL (full methods) consistently achieve the highest power, outperforming all ablated variants at every sample size. Removing either component leads to a drop in power. Importantly, both of these variants still outperform the plain aggregation baseline. This indicates that each component—diversity in the kernel choices and selection inference during testing—independently contributes to improving test power. Their combination in DUAL has a cumulative effect, yielding the best performance overall.

**Kernel Selection Probability by Selection Inference.** We next examine how the selection inference behaves under the null and alternative through the kernel selection probabilities. Under the null hypothesis (see Figure 4 (b) and (f)), each kernel is selected with roughly 50% probability, which means the selection is approximately uniform across the pool of diverse kernels and guarantee that selection inference has *no* influence on the control of Type-I error. This holds consistently across sample sizes, indicating that in the absence of a signal the procedure effectively randomizes over the kernel choices without bias. In contrast, under the alternative hypothesis (see Figure 4 (c) and

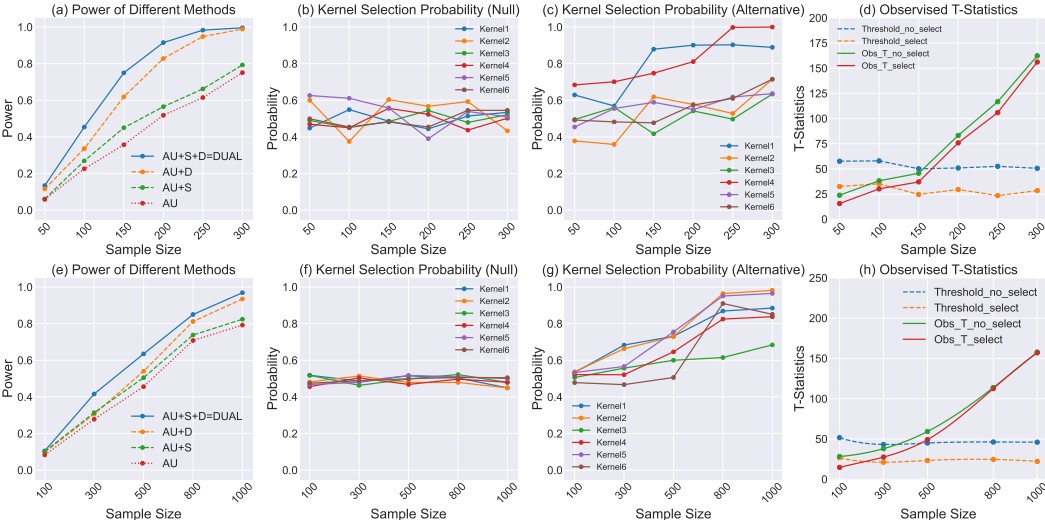

**Figure 4:** Ablation Study on the effectiveness of diversity and selection inference. $(a-d)$ are ablation study for MMD-DUAL; $(e-h)$ are ablation study for HSIC-DUAL. $(a, e)$ Test power for model variants: AU represents simple Aggregated $U$-Statistics; S represents selection inference technique; D represents considering diversity into AU; AU+S+D refers to our proposed DUAL.

(g)), the selection probabilities become increasingly concentrate on the most informative kernels as the sample size grows. In other words, the selection inference technique correctly detects which kernel is capturing the existing dependency or distribution difference, and it selects that kernel with ever-growing frequency. For instance, on the Higgs dataset (Figure 4 (g)), the kernels (Kernel 2 and Kernel 6) that best capture the dependence structure are chosen far more often than others, with its selection probability rising well above 0.5 and eventually approaching nearly 1.0 as the sample size increases. This trend demonstrates that the procedure progressively focuses on the kernels that yield the strongest test statistic, effectively leveraging the most useful features of the data.

**Testing Threshold and Observed Test Statistics.** We now analyze the impact of selection inference on the test power, which mainly depends on the testing threshold and the observed test statistic (see Figure 4 (d) and (h)). From the two dotted lines, we observe that the testing threshold (derived by $1 - \alpha$ quantiles of the aggregated statistic under $H_0$ or wild bootstrap) of applying selection inference is approximately 50% that of the aggregated statistic on the original full kernel set, because any given kernel is included in the test statistic only about 50% of the time on average in selection inference procedure. Furthermore, from the two solid lines, we can find that the observed test statistic produced by DUAL is initially approximately half that of the no-selection counterpart for small sample sizes. This is expected: at lower sample size, the selection inference might not able to determine which kernels are signal-carrying, so its test statistic starts off lower. However, as the sample size increases, the selected kernel is almost always the most informative one, and thus the adaptive test statistic grows and eventually matches the magnitude of the no-selection test statistic. Crucially, throughout this process, the DUAL enjoys the advantage of a lower threshold, and as sample size increases, its statistic catches up to the no-selection statistic. This combination—a growing test statistic that converges to the no-selection level, together with a consistently reduced critical threshold—means that DUAL achieves higher power at all sample sizes.

In summary, the ablation results show that both diversity and selection inference contribute to performance gains, and the selection inference mechanism not only identifies informative kernels under $H_1$ but also effectively control the Type-I error under $H_0$, leading to a substantial improvement in test power for the full DUAL method. A more detailed analytical explanation and analysis to support the results in Figure 4 is provided in Example 1 (Appendix D).

### 6.3 Computational and Scalability Analysis

**Time Complexity Analysis.** Table 1 summarizes the time complexity of MMD-DUAL[10] in both the training and testing phases, and compares it with previous methods, MMDAgg and MMD-FUSE.

---

[10]HSIC-DUAL exhibits the same time complexity, as both follow the formulation of the second-order U-statistic described in Section 2.

**Table 1:** Time complexity of MMD-DUAL, MMDAgg and MMD-FUSE in training phase (using Adam optimizer [64]) and testing phase (using wild bootstrap)

| Time Complexity | MMD-DUAL | MMDAgg | MMD-FUSE |
|---|---|---|---|
| Training | $\mathcal{O}((n^2c^2 + c^3 + T) * M)$ | N/A | N/A |
| Testing | $\mathcal{O}(Bcn^2 + n^2c^2 + c^3)^{11}$ | $\mathcal{O}(n^2c(B + B'))$ | $O(n^2cB)$ |

MMDAgg and MMD-FUSE select the kernel parameters using heuristic methods without a training procedure. For all methods, the complexity is dominated by the quadratic term in the sample size, i.e., $\mathcal{O}(n^2)$, which corresponds to the computational cost of computing the second-order U-statistic. Other factors, including the number of kernels $c$, the number of optimization parameters $T$, the number of optimization epochs $M$, the number of wild bootstrap iterations $B$, and the number of permutation tests $B'$, are treated as constants that do not scale with $n$.

**Scalability.** Regarding the scalability of DUAL method, the time complexity are quadratic or cubic related to the size of kernel pool. In that way, using a very large candidate kernel pool may lead to computational inefficiency. In practical implementations of two-sample and independence testing, the kernel set is initialized using heuristic methods, following the methodology of [7–9]. As shown in these studies, increasing the number of kernels beyond a moderate size does not yield noticeable improvements in test power. Specifically, as shown in Figure 6 in Section 5.7 of [8], increasing the number of kernels from 10 to 100 and even to 1,000 does not improve power. This supports the choice of using a small number of kernels (e.g.,$c = 10$), as there is nothing to gain empirically from using a finer discretization for the bandwidths of kernels. In fact, the referenced study even considers aggregating 12,000 kernels.

For high-dimensional data, kernel methods scale linearly with the input dimension, since the computation of each pairwise kernel evaluation (e.g., in Gaussian kernel) involves an inner product. This cost arises only during the construction of the kernel matrix (typically through pairwise distance computations), which can be efficiently implemented and is rarely the bottleneck in practice.

## 7    Conclusions

In this paper, we identify that kernel selection markedly influences the performance of aggregated-kernel two-sample and independence tests. To address the challenge of optimizing kernel aggregation for nonparametric hypothesis testing, we introduce *Diverse U-statistic Aggregation with Learned Kernels* (DUAL). Through analysis, we demonstrate the importance of balancing kernel diversity with individual-kernel effectiveness to enhance statistical power. Our method integrates kernel selection via covariance-informed diversity measures, thereby mitigating the adverse effects of redundant or weak kernels. Experiments on a variety of benchmarks show that DUAL-based tests (MMD-DUAL and HSIC-DUAL) consistently outperform state-of-the-art approaches, confirming their effectiveness. Future work will extend this diversity-driven aggregation approach to advanced ensemble-regularized optimization techniques and will explore its effectiveness on broader classes of $U$-statistics across diverse tasks, e.g., goodness-of-fit testing. Our method of selection inference for the kernel collection could be generalised to also be applicable to adaptive multiple testing.

## Acknowledgments and Disclosure of Funding

This research was supported by The University of Melbourne's Research Computing Services and the Petascale Campus Initiative. ZJZ, XYT and CL are supported by the Melbourne Research Scholarship. ZJZ and XYT are also supported by the ARC with grant number DE240101089. LHP is supported by the ARC with grant number LP240100101. AS is supported by a UKRI Turing AI World-Leading Researcher Fellowship with grant number G111021. DJS is supported in part by the Canada CIFAR AI Chairs program. FL is supported by the ARC with grant number DE240101089, LP240100101, DP230101540 and the NSF&CSIRO Responsible AI program with grant number 2303037.

---

[11]This differs from the time complexity reported in the rebuttal, which was $\mathcal{O}((n^2c^2 + c^3)B)$. The actual complexity can be lower, as the covariance matrix $\widehat{\Sigma}_{H_0}$ needs to be computed only once in the testing procedure.

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

# Appendix

## A  Asymptotic Theory for the Proposed Statistics

### A.1  Asymptotic Behavior of the Aggregated Statistic

In this section, we investigate the asymptotic behaviors of various statistics defined in Section 4. To maintain notational consistency, we compute the statistics over the sample $W = \{\boldsymbol{w}_i\}_{i=1}^n$ (which is independent of the kernel set $\mathcal{K}$), as done in Section 4. However, these results remain valid for the testing samples $W_{te}$ discussed in next Section A.2, since $W_{te}$ is also independent of the kernel set $\mathcal{K}$.

We first present the asymptotic behavior of the multivariate $U$-statistics with multiple kernels, i.e., $n \cdot \boldsymbol{U}_n^{\mathcal{K}}(W)$, as follows.

**Theorem 2.** *Let $\mathcal{K} = \{\kappa_1, \kappa_2, ..., \kappa_c\}$ be a set of $c$ kernels such that $\kappa_i$ with $i \in [c]$ is charateristic and bounded. Then, under null hypothesis $H_0$, for first-order degenerate U-statistic, we have*

$$n \cdot \boldsymbol{U}_n^{\mathcal{K}}(W) \xrightarrow{d} G_{\mathcal{K}} = (I_2(h(\cdot; \kappa_1)), I_2(h(\cdot; \kappa_2)), ..., I_2(h(\cdot; \kappa_c)))^T \ ,$$

*where $I_2(\cdot)$ is the multiple Wiener-Itô integral (Definition 10, Appendix B.2). Furthermore, the characteristic function of $G_{\mathcal{K}}$ evaluated at $\boldsymbol{\eta} = (\eta_1, ..., \eta_c)^T \in \mathbb{R}^c$ is defined as*

$$\Phi(\boldsymbol{\eta}) = E\left[e^{\iota \boldsymbol{\eta}^T G_{\mathcal{K}}}\right] = \prod_{\nu=1}^{\infty} \frac{\exp(-\iota \lambda_\nu)}{\sqrt{1 - 2\iota \lambda_\nu}} \ ,$$

*where $\{\lambda_\nu\}_{\nu=1}^{\infty}$ are eigenvalues of the Hilbert-Schmidt operator $\mathcal{H}_{\mathcal{K}_{\boldsymbol{\eta}}} : L^2(\mathcal{W}, \mathbb{W}) \to L^2(\mathcal{W}, \mathbb{W})$ as*

$$\mathcal{H}_{\mathcal{K}_{\boldsymbol{\eta}}}[f](\boldsymbol{w}_1) = \int_{-\infty}^{\infty} h(\boldsymbol{w}_1, \boldsymbol{w}_2; \mathcal{K}_{\boldsymbol{\eta}}) f(\boldsymbol{w}_2) d\mathbb{W}(\boldsymbol{w}_2) \ ,$$

*where $\mathcal{K}_{\boldsymbol{\eta}}(\cdot, \cdot) = \sum_{j=1}^c \eta_j \kappa_j(\cdot, \cdot)$ and $h(\boldsymbol{w}_1, \boldsymbol{w}_2; \mathcal{K}_{\boldsymbol{\eta}}) = \sum_{j=1}^c \eta_j h(\boldsymbol{w}_1, \boldsymbol{w}_2; \kappa_j)$.*

For the covariance matrix of $n \cdot \boldsymbol{U}_n^{\mathcal{K}}(W)$ under $H_0$, i.e., $\Sigma_{H_0}$, we present its asymptotic behavior as follows.

**Lemma 3.** *For a bounded function $h(\cdot; \kappa)$ with kernel $\kappa$, the estimator $\widehat{\Sigma}_{H_0}$ defined in Eqn. (3) satisfies $\widehat{\Sigma}_{H_0} \xrightarrow{p} \Sigma_{H_0}$.*

Building on Theorem 2 and Lemma 3, the asymptotic behavior of $T_n^{\mathcal{K}}(W)$ is established in the following corollary using Slutsky's theorem [65].

**Corollary 4.** *Under the null hypothesis $H_0$, the statistic satisfies $T_n^{\mathcal{K}}(W) \xrightarrow{d} G_{\mathcal{K}}^T \Sigma_{H_0}^{-1} G_{\mathcal{K}}$.*

Subsequently, we establish that under the alternative hypothesis $H_1$, our ensemble statistics asymptotically converge in distribution to a normal law.

**Theorem 5.** *Under the alternative hypothesis $H_1$, for non-degenerate function $h(\cdot; \kappa)$ with $\kappa \in \mathcal{K}$, and assuming $E[h^2(\boldsymbol{w}_1, \boldsymbol{w}_2; \kappa)] < \infty$ for each $\kappa \in \mathcal{K}$, the following holds*

$$\sqrt{n}(\boldsymbol{U}_n^{\mathcal{K}}(W) - \boldsymbol{U}^{\mathcal{K}}(\mathbb{W})) \xrightarrow{d} \mathcal{N}(\boldsymbol{0}, \Sigma_{H_1}) \ ,$$

*where $\boldsymbol{U}^{\mathcal{K}}(\mathbb{W}) = E\left[\boldsymbol{U}_n^{\mathcal{K}}(W)\right]$ and $\Sigma_{H_1}$ is the covariance matrix of $\sqrt{n}\boldsymbol{U}_n^{\mathcal{K}}(W)$ under $H_1$, whose entries consist of (co)variances $\sigma_{H_1,a,b}$ with $1 \leq a, b \leq c$ defined as*

$$\sigma_{H_1,a,b} = 4\left(E\left[h_1(\boldsymbol{w}_1; \kappa_a)h_1(\boldsymbol{w}_1; \kappa_b)\right] - U^{\kappa_a}(\mathbb{W})U^{\kappa_b}(\mathbb{W})\right) \ ,$$

*where $U^{\kappa}(\mathbb{W}) = E[U_n^{\kappa}(W)]$. Furthermore, the asymptotic distribution of $T_n^{\mathcal{K}}(W)$ is given by*

$$n^{-3/2}\left(T_n^{\mathcal{K}}(W) - n^2\left(\boldsymbol{U}^{\mathcal{K}}(\mathbb{W})\right)^T \widehat{\Sigma}_{H_0}^{-1} \boldsymbol{U}^{\mathcal{K}}(\mathbb{W})\right) \xrightarrow{d} \mathcal{N}(0, \sigma_{H_1}^2) \ ,$$

*where $\sigma_{H_1}^2 = 4\left(\boldsymbol{U}^{\mathcal{K}}(\mathbb{W})\right)^T \Sigma_{H_0}^{-1} \Sigma_{H_1} \Sigma_{H_0}^{-1} \boldsymbol{U}^{\mathcal{K}}(\mathbb{W})$.*

## A.2   Asymptotic Behavior of the Test Statistic with Selection Inference

In this section, we analyze the asymptotic behavior of the test statistics under selection inference, as introduced in Section 5.2. We regard that the kernel set $\mathcal{K}$ and the signum vector $\boldsymbol{F}_{tr}$ are fixed, as they are independent of the testing sample $W_{te}$ on which the statistics are computed. Accordingly, the sample size $n$ refers exclusively to the size of the testing sample. Throughout this section, we compute $\widehat{\Sigma}_{H_0}$ and $\widehat{\boldsymbol{L}}_{H_0}$ based on $W_{te}$, i.e., $\widehat{\Sigma}_{H_0} = \widehat{\Sigma}_{H_0}(W_{te})$ and $\widehat{\boldsymbol{L}}_{H_0} = \widehat{\boldsymbol{L}}_{H_0}(W_{te})$. For notational convenience, we omit the explicit dependence on $W_{te}$.

We first write the test statistic $\mathcal{T} = \| \max(n \cdot \operatorname{diag}(\boldsymbol{F}_{tr})\widehat{\boldsymbol{L}}_{H_0}^{-1}\boldsymbol{U}_n^{\mathcal{K}}(W_{te}), \boldsymbol{0})\|_2^2$ according to the definition in Eqn. (9). By Lemma 3, we have that $\widehat{\boldsymbol{L}}_{H_0} \xrightarrow{p} \boldsymbol{L}_{H_0}$ with $\Sigma_{H_0} = \boldsymbol{L}_{H_0}\boldsymbol{L}_{H_0}$ based on the same Schur decomposition [52]. In the following theorem, we present the asymptotic behavior of our test under the null hypothesis $H_0$.

**Theorem 6.** *Under null hypothesis $H_0$, both the test statistic $\mathcal{T}$ and the wild bootstrap statistic $\mathcal{T}^b$ ($b \in [B]$) for MMD and HSIC converge in distribution to $\| \max\left(\operatorname{diag}(\boldsymbol{F}_{tr})\boldsymbol{L}_{H_0}^{-1}G_{\mathcal{K}}, \boldsymbol{0}\right)\|_2^2$, where $G_{\mathcal{K}} = (I_2(h(\cdot; \kappa_1)), I_2(h(\cdot; \kappa_2)), ..., I_2(h(\cdot; \kappa_c)))^T$.*

Having established the validity of our test under the null hypothesis, we now investigate the asymptotic behavior under the alternative hypothesis $H_1$, where the vector $\operatorname{diag}(\boldsymbol{F}_{tr})\widehat{\boldsymbol{L}}_{H_0}^{-1}\boldsymbol{U}_n^{\mathcal{K}}(W_{te})$ converges in distribution to a normal law, as a consequence of Theorem 5 and $\widehat{\boldsymbol{L}}_{H_0} \xrightarrow{p} \boldsymbol{L}_{H_0}$ by Lemma 3.

**Corollary 7.** *Under alternative hypothesis $H_1$, the following asymptotic distribution holds*

$$\sqrt{n} \cdot \operatorname{diag}(\boldsymbol{F}_{tr})\widehat{\boldsymbol{L}}_{H_0}^{-1}\boldsymbol{U}_n^{\mathcal{K}}(W_{te}) \xrightarrow{d}$$
$$\mathcal{N}\left(\sqrt{n} \cdot \operatorname{diag}(\boldsymbol{F}_{tr})\boldsymbol{L}_{H_0}^{-1}\boldsymbol{U}^{\mathcal{K}}(\mathbb{W}), \operatorname{diag}(\boldsymbol{F}_{tr})\boldsymbol{L}_{H_0}^{-1}\Sigma_{H_1}\boldsymbol{L}_{H_0}^{-1}\operatorname{diag}(\boldsymbol{F}_{tr})\right) .$$

# B   Detailed Proofs for Our Theoretical Results

## B.1   The Detailed Proofs of Theorem 1

We begin with an assumption as follows.

**Assumption 1.** *Under alternative hypothesis $H_1$, given the signum vector $\boldsymbol{F}_{tr}$, we assume that there exists at least one index $i \in \{1, 2, ..., c\}$ such that*

$$F_{tr,i} = \operatorname{sgn}\left(\boldsymbol{L}_{H_0}^{-1}\boldsymbol{U}^{\mathcal{K}}(\mathbb{W})\right)_i ,$$

*where $\mathbf{a}_i$ indicates the $i$-th coordinate of vector $\mathbf{a}$, $\boldsymbol{U}^{\mathcal{K}}(\mathbb{W}) = E\left[\boldsymbol{U}_n^{\mathcal{K}}(W_{te})\right]$, and $\Sigma_{H_0} = \boldsymbol{L}_{H_0}\boldsymbol{L}_{H_0}$ is the same Schur decomposition [52] applied to $\widehat{\Sigma}_{H_0} = \widehat{\boldsymbol{L}}_{H_0}\widehat{\boldsymbol{L}}_{H_0}$, with $\widehat{\Sigma}_{H_0} \xrightarrow{p} \Sigma_{H_0}$ by Lemma 3.*

This assumption requires that the estimated signum vector, defined as

$$\boldsymbol{F}_{tr} = \operatorname{sgn}\left(\widehat{\boldsymbol{L}}_{H_0}^{-1}\boldsymbol{U}_n^{\mathcal{K}}(W_{tr})\right) \in \{-1, +1\}^c ,$$

matches the ground-truth vector $\operatorname{sgn}\left(\boldsymbol{L}_{H_0}^{-1}\boldsymbol{U}^{\mathcal{K}}(\mathbb{W})\right)$ in at least one coordinate.

Notably, throughout the statement and proof of Theorem 1, we treat $\boldsymbol{F}_{tr}$ as fixed, since it is derived from the training samples and is independent of the testing samples analyzed here. We present the detailed proofs of Theorem 1 as follows.

*Proof.* We first prove that, under the null hypothesis $H_0$, the test in Eqn.(11) has type-I error bounded by $\alpha$, i.e., $\operatorname{Pr}_{H_0}(\mathfrak{h}(X, Y; \kappa) = 1) \leq \alpha$, which holds non-asymptotically.

Given the signum vector $\boldsymbol{F}_{tr}$, we write the test statistic with selection inference as follows

$$\mathcal{T} = \| \max(n \cdot \operatorname{diag}(\boldsymbol{F}_{tr})\widehat{\boldsymbol{L}}_{H_0}^{-1}\boldsymbol{U}_n^{\mathcal{K}}(W_{te}), \boldsymbol{0})\|_2^2 , \tag{12}$$

according to the definition in Eqn. (9). In a similar manner, we can write the $b$-th wild bootstrap statistic as

$$\mathcal{T}^b = \left\|\max\left(n \cdot \operatorname{diag}(\boldsymbol{F}_{tr})\widehat{\boldsymbol{L}}_{H_0}^{-1}\boldsymbol{U}_n^{\mathcal{K},b}(W_{te}), \boldsymbol{0}\right)\right\|_2^2 .$$

Building on the results of [7, Appendix F.1], the statistics $\boldsymbol{U}_n^{\mathcal{K},1}(W_{te}), \boldsymbol{U}_n^{\mathcal{K},2}(W_{te}), \ldots, \boldsymbol{U}_n^{\mathcal{K},B}(W_{te})$, along with the original statistic $\boldsymbol{U}_n^{\mathcal{K}}(W_{te})$, are exchangeable under the null hypothesis for both the two-sample and independence testing problems. By combining the exchangeability and [66, Theorem 1], it follows that the wild bootstrap statistics $\mathcal{T}^1, \mathcal{T}^2, \ldots, \mathcal{T}^B$ and the original test statistic $\mathcal{T}$ are likewise exchangeable under the null hypothesis. Consequently, by applying the exchangeability-based argument of [67, Lemma 1], the test defined in Eqn. (11) achieves non-asymptotic control of the type-I error at level $\alpha$ under the null hypothesis, i.e.,

$$\Pr_{H_0}(\mathfrak{h}(X, Y; \kappa) = 1) \leq \alpha .$$

Having established the control of type-I error, we now proceed to analyze the asymptotic behavior of $\mathcal{T}$ under the null hypothesis, as a preparatory step for proving the consistency of the test power under the alternative hypothesis. Specifically, under the null hypothesis $H_0$, by invoking the large-deviation bound for $U$-statistic (Theorem 12), the following joint convergence in probability holds

$$\boldsymbol{U}_n^{\mathcal{K}}(W_{te}) \xrightarrow{p} E\left[\boldsymbol{U}_n^{\mathcal{K}}(W)\right] \qquad \text{with} \qquad E\left[\boldsymbol{U}_n^{\mathcal{K}}(W)\right] = \boldsymbol{0} . \tag{13}$$

By Lemma 3, we have that $\widehat{\Sigma}_{H_0} \xrightarrow{p} \Sigma_{H_0}$. Based on the same Schur decomposition [52], we denote by $\widehat{\Sigma}_{H_0} = \widehat{\boldsymbol{L}}_{H_0}\widehat{\boldsymbol{L}}_{H_0}$ and $\Sigma_{H_0} = \boldsymbol{L}_{H_0}\boldsymbol{L}_{H_0}$. The continuous-mapping theorem [68] then yields that $\widehat{\boldsymbol{L}}_{H_0} \xrightarrow{p} \boldsymbol{L}_{H_0}$. Combining the two convergences in probability of $\boldsymbol{U}_n^{\mathcal{K}}(W_{te})$ and $\widehat{\boldsymbol{L}}_{H_0}$, for the test statistic in Eqn. (12), it follows that

$$\frac{\mathcal{T}}{n} \xrightarrow{p} \left\|\max\left(\mathrm{diag}(\boldsymbol{F}_{tr})\, \boldsymbol{L}_{H_0}^{-1}\, \boldsymbol{0}, \boldsymbol{0}\right)\right\|_2^2 = 0 ,$$

by continuous-mapping theorem. Since the wild bootstrap statistics $\mathcal{T}^b$ with $b \in [B]$ and the original test statistic $\mathcal{T}$ are exchangeable under the null hypothesis, if follows that

$$\frac{\mathcal{T}^b}{n} \xrightarrow{p} \left\|\max\left(\mathrm{diag}(\boldsymbol{F}_{tr})\, \boldsymbol{L}_{H_0}^{-1}\, \boldsymbol{0}, \boldsymbol{0}\right)\right\|_2^2 = 0 . \tag{14}$$

Next, we prove the consistency of the test power **under the alternative hypothesis** $H_1$. Similarly, by invoking the large-deviation bound for $U$-statistic (Theorem 12), it follows that

$$\boldsymbol{U}_n^{\mathcal{K}}(W_{te}) \xrightarrow{p} \boldsymbol{U}^{\mathcal{K}}(\mathbb{W}) \qquad \text{with} \qquad \boldsymbol{U}^{\mathcal{K}}(\mathbb{W}) = E\left[\boldsymbol{U}_n^{\mathcal{K}}(W)\right] , \tag{15}$$

where each dimension of $\boldsymbol{U}^{\mathcal{K}}(\mathbb{W})$ is strictly positive for MMD [3, Lemma 1] and HSIC [69, Theorem 6] statistics in two-sample and independence testing problems with characteristic kernels. Similarly, combined with the results of Eqns. (12) and (15), and $\widehat{\boldsymbol{L}}_{H_0} \xrightarrow{p} \boldsymbol{L}_{H_0}$, it follows that

$$\frac{\mathcal{T}}{n} \xrightarrow{p} \|\max(\mathrm{diag}(\boldsymbol{F}_{tr})\boldsymbol{L}_{H_0}^{-1}\boldsymbol{U}^{\mathcal{K}}(\mathbb{W}), \boldsymbol{0})\|_2^2 . \tag{16}$$

Based on the Assumption 1, the positive-definiteness of $\boldsymbol{L}_{H_0}$ and the strictly positive $\boldsymbol{U}^{\mathcal{K}}(\mathbb{W})$, we have that there exists at least one index $i \in \{1, 2, ..., c\}$ and a constant $C_1 > 0$ such that

$$\left(\mathrm{diag}(\boldsymbol{F}_{tr})\boldsymbol{L}_{H_0}^{-1}\boldsymbol{U}^{\mathcal{K}}(\mathbb{W})\right)_i = C_1 .$$

Then, combined with Eqn. (16), it follows that there exists a constant $C_2 \geq C_1 > 0$ such that

$$\frac{\mathcal{T}}{n} \xrightarrow{p} C_2 . \tag{17}$$

Given Eqns. (14) and (17), it follows that

$$\frac{\mathcal{T}}{n} - \frac{\mathcal{T}^b}{n} \xrightarrow{p} C_2 - 0 = C_2 > 0 ,$$

by Slutsky's Theorem [65].

By the definition of convergence in probability, for any $\varepsilon \in (0, C_2)$,

$$\Pr_{H_1}\left(\left|\frac{\mathcal{T} - \mathcal{T}^b}{n} - C_2\right| < \varepsilon\right) \to 1 \quad \implies \quad \Pr_{H_1}\left(\frac{\mathcal{T} - \mathcal{T}^b}{n} > C_2 - \varepsilon\right) \to 1 .$$

Taking $\varepsilon = C_2/2$ gives $C_2 - \varepsilon = C_2/2 > 0$, so

$$\mathrm{Pr}_{H_1}\left((\mathcal{T} - \mathcal{T}^b)/n > 0\right) \geq \mathrm{Pr}_{H_1}\left((\mathcal{T} - \mathcal{T}^b)/n > C_2/2\right) \to 1,$$

and hence

$$\lim_{n\to\infty} \mathrm{Pr}_{H_1}(\mathcal{T} > \mathcal{T}^b) = 1 . \tag{18}$$

Building on the results of [7, Appendix F.1], the wild bootstrap is equivalent to applying a subgroup of permutations. Then, for the test in Eqn.(11), the result of Eqn. (18) guarantees that the sufficient condition for consistency of [70, Lemma 8] on permutation test is satisfied, implying that

$$\lim_{n\to\infty} \mathrm{Pr}_{H_1}(\mathfrak{h}(X, Y; \kappa) = 1) = 1 .$$

This completes the proof. $\qquad\square$

## B.2 The Detailed Proofs of Theorem 2

The proof of Theorem 2 follows the proof of [28, Theorem 3.1], with some modifications to accommodate our framework. We begin with some useful Definitions below.

**Definition 8.** *[28, Definition 3.1] A Gaussian stochastic measure on $(\mathcal{W}, \mathscr{B}(\mathcal{W}), \mathbb{W})$ is a collection of random variables $\{\mathcal{Z}_{\mathbb{W}}(A) : A \in \mathscr{B}(\mathcal{W})\}$ defined on a probability space $(\Omega, \mathcal{F}, \mu)$ such that the following holds*

- *$\mathcal{Z}_{\mathbb{W}}(A) \sim \mathcal{N}(0, \mathbb{W}(A))$, for all $A \in \mathscr{B}(\mathcal{W})$.*

- *For any finite collection of disjoint sets $A_1, ..., A_t \in \mathscr{B}(\mathcal{W})$, the random variables $\{\mathcal{Z}_{\mathbb{W}}(A_1), \mathcal{Z}_{\mathbb{W}}(A_2), ..., \mathcal{Z}_{\mathbb{W}}(A_t)\}$ are independent and*

$$\mathcal{Z}_{\mathbb{W}}(\cup_{s=1}^t A_s) = \sum_{s=1}^t \mathcal{Z}_{\mathbb{W}}(A_s) .$$

Let $L^2(\mathcal{W}^2, \mathscr{B}(\mathcal{W}^2), \mathbb{W}^2)$ be the space of measurable functions $f : \mathcal{W}^2 \to \mathbb{R}$ and

$$\|f\|^2 = \int_{\mathcal{W}^2} |f(\boldsymbol{w}_1, \boldsymbol{w}_2)|^2 d\mathbb{W}(\boldsymbol{w}_1) d\mathbb{W}(\boldsymbol{w}_2) < \infty .$$

Furthermore, we define $\mathcal{E}_2 \subseteq L^2(\mathcal{W}^2, \mathscr{B}(\mathcal{W}^2), \mathbb{W}^2)$ as the set of all elementary functions as

$$f(t_1, t_2) = \sum_{1 \leq i_1, i_2 \leq n} a_{i_1,i_2} \mathbf{1}\{(t_1, t_2) \in A_{i_1} \times A_{i_2}\} ,$$

where $A_1, ..., A_n \in \mathscr{B}(\mathcal{W})$ are pairwise disjoint and $a_{i_1,i_2}$ is the coefficient of the elementary function, which is zero if two indices are equal. The multiple Weiner-Itô integral for the functions in $\mathcal{E}_2$ is defined as follows.

**Definition 9.** *[28, Definition 3.2] The $m$-dimensional Weiner-Itô stochastic integral, with respect to the Gaussian stochastic measure $\{\mathcal{Z}_{\mathbb{W}}(A), A \in \mathscr{B}(\mathcal{W})\}$, for the function $f \in \mathcal{E}_2$ is defined as*

$$\begin{aligned} I_2^e(f) &= \int_{\mathcal{W}^2} f(\boldsymbol{w}_1, \boldsymbol{w}_2) d\mathcal{Z}_{\mathbb{W}}(\boldsymbol{w}_1) d\mathcal{Z}_{\mathbb{W}}(\boldsymbol{w}_2) \\ &= \sum_{1 \leq i_1, i_2 \leq n} a_{i_1,i_2} \mathcal{Z}_{\mathbb{W}}(A_{i_1}) \times \mathcal{Z}(A_{i_2}) . \end{aligned}$$

By taking limits over the set $\mathcal{E}_2$, we can extend the above multiple Weiner-Itô integral for elementary functions to functions in $L^2(\mathcal{W}^2, \mathscr{B}(\mathcal{W}^2), \mathbb{W}^2)$ as follows.

**Definition 10.** *[28, Multiple Weiner-Itô integral for general $L_2$-functions, Definition 3.3] The 2-dimensional Weiner-Itô stochastic integral for a function $f \in L^2(\mathcal{W}^2, \mathscr{B}(\mathcal{W}^2), \mathbb{W}^2)$ is defined as the $L_2$ limit of the sequence $\{I_2(f_\ell)\}_{\ell \geq 1}$, where $\{f_\ell\}_{\ell \geq 1}$ is a sequence such that $f_\ell \in \mathcal{E}_2$ with $\lim_{\ell\to\infty} \|f_\ell - f\| = 0$. This is denoted by*

$$I_2(f) = \int_{\mathcal{W}^2} f(\boldsymbol{w}_1, \boldsymbol{w}_2) d\mathcal{Z}_{\mathbb{W}}(\boldsymbol{w}_1) d\mathcal{Z}_{\mathbb{W}}(\boldsymbol{w}_2) .$$

Given the Multiple Weiner-Itô integral for general $L_2$-functions. **We now present the detailed proofs of Theorem 2 as follows.**

*Proof.* Recall the definition of $\boldsymbol{U}_n^{\mathcal{K}}(W)$ in Eqn. (2), we have that for $\boldsymbol{\eta} = (\eta_1, \eta_2, ..., \eta_c)^T \in \mathbb{R}^c$,

$$
\begin{aligned}
\boldsymbol{\eta}^T \boldsymbol{U}_n^{\mathcal{K}}(W) &= \sum_{j=1}^{c} \eta_j U_n^{\kappa_j}(W) \\
&= \binom{n}{2}^{-1} \sum_{j=1}^{c} \eta_j \sum_{1 \le i_1 < i_2 \le n} h(\boldsymbol{w}_{i_1}, \boldsymbol{w}_{i_2}; \kappa_j) \\
&= \binom{n}{2}^{-1} \sum_{1 \le i_1 < i_2 \le n} \sum_{j=1}^{c} \eta_j h(\boldsymbol{w}_{i_1}, \boldsymbol{w}_{i_2}; \kappa_j) \\
&= \binom{n}{2}^{-1} \sum_{1 \le i_1 < i_2 \le n} h(\boldsymbol{w}_{i_1}, \boldsymbol{w}_{i_2}; \mathcal{K}_{\boldsymbol{\eta}}) ,
\end{aligned}
$$

where

$$
h(\boldsymbol{w}_{i_1}, \boldsymbol{w}_{i_2}; \mathcal{K}_{\boldsymbol{\eta}}) = \sum_{j=1}^{c} \eta_j h(\boldsymbol{w}_{i_1}, \boldsymbol{w}_{i_2}; \kappa_j) .
$$

It is easy to see that the function $h(\cdot; \mathcal{K}_{\boldsymbol{\eta}})$ is a measurable and symmetric function, and $h(\cdot; \mathcal{K}_{\boldsymbol{\eta}}) \in L^2(\mathcal{W}^2, \mathscr{B}(\mathcal{W}^2), \mathbb{W}^2)$ based on bounded and characteristic kernels. Moreover, the function $h(\cdot; \mathcal{K}_{\boldsymbol{\eta}})$ is first-order degenerate, as this property is inherited from the first-order degeneracy of each component function $h(\cdot; \kappa)$ with $\kappa \in \mathcal{K}$. Then, by [71, Corollary 4.4.2, Section 4.4], we obtain

$$
n \cdot \boldsymbol{\eta}^T \boldsymbol{U}_n^{\mathcal{K}}(W) \overset{d}{\to} \sum_{\nu=1}^{\infty} \lambda_\nu(Z_\nu^2 - 1) , \tag{19}
$$

where the $Z_\nu$ are i.i.d. random variables drawn from $\mathcal{N}(0, 1)$ and the $\{\lambda_\nu\}_{\nu=1}^{\infty}$ are the eigenvalues of the Hilbert-Schmidt operator $\mathcal{H}_{\mathcal{K}_{\boldsymbol{\eta}}} : L^2(\mathcal{W}, \mathscr{B}(\mathcal{W}), \mathbb{W}) \to L^2(\mathcal{W}, \mathscr{B}(\mathcal{W}), \mathbb{W})$ defined as

$$
\mathcal{H}_{\mathcal{K}_{\boldsymbol{\eta}}}[f](\boldsymbol{w}_1) = \int_{-\infty}^{\infty} h(\boldsymbol{w}_1, \boldsymbol{w}_2; \mathcal{K}_{\boldsymbol{\eta}}) f(\boldsymbol{w}_2) d\mathbb{W}(\boldsymbol{w}_2) . \tag{20}
$$

By [72, Theorem 6.1, Section 6], we have the characteristic function $\Phi(\boldsymbol{\eta})$ as follows

$$
\begin{aligned}
E\left[\exp\left(\iota \cdot n \cdot \boldsymbol{\eta}^T \boldsymbol{U}_n^{\mathcal{K}}(W)\right)\right] &\overset{d}{\to} E\left[\exp\left(\iota \cdot \sum_{\nu=1}^{\infty} \lambda_\nu(Z_\nu^2 - 1)\right)\right] \\
&= \prod_{\nu=1}^{\infty} \frac{\exp(-\iota\lambda_\nu)}{\sqrt{1 - 2\iota\lambda_\nu}} = \Phi(\boldsymbol{\eta}) .
\end{aligned} \tag{21}
$$

Here, $\boldsymbol{\eta} \in \mathbb{R}^c$ can be chosen arbitrarily and Lemma 11 establishes the continuity of $\Phi(\boldsymbol{\eta})$ at $\boldsymbol{\eta} = \boldsymbol{0} \in \mathbb{R}^c$. By Lévy's Continuity Theorem [73, Theorem 3.3.17], a random vector $Z_{\mathcal{K}}$ exists with characteristic function $\Phi(\boldsymbol{\eta})$ such that

$$
n \cdot \boldsymbol{U}_n^{\mathcal{K}}(W) \overset{d}{\to} Z_{\mathcal{K}} .
$$

Next, we establish that $Z_{\mathcal{K}}$ can be represented as $G_{\mathcal{K}}$. To achieve this, we leverage the linearity property of multiple stochastic integrals, which allows us to write the characteristic function of $G_{\mathcal{K}}$ at

$\boldsymbol{\eta}$ as follows

$$
\begin{aligned}
\Phi'(\boldsymbol{\eta}) &= E\left[\exp\left(\iota\boldsymbol{\eta}^T G_{\mathcal{K}}\right)\right] \\
&= E\left[\exp\left(\iota\sum_{j=1}^{c}\eta_j I_2(h(\cdot;\kappa_j))\right)\right] \\
&= E\left[\exp\left(\iota I_2\left(\sum_{j=1}^{c}\eta_j h(\cdot;\kappa_j)\right)\right)\right] \\
&\overset{(a)}{=} E\left[\exp\left(\iota I_2\left(h(\cdot;\mathcal{K}_{\boldsymbol{\eta}})\right)\right)\right] \\
&\overset{(b)}{=} \prod_{\nu=1}^{\infty}\frac{\exp(-\iota\lambda_\nu')}{\sqrt{1-2\iota\lambda_\nu'}}
\end{aligned}
$$

where equality $(a)$ holds for $h\left(\boldsymbol{w}_{i_1},\boldsymbol{w}_{i_2};\mathcal{K}_{\boldsymbol{\eta}}\right) = \sum_{j=1}^{c}\eta_j h(\boldsymbol{w}_{i_1},\boldsymbol{w}_{i_2};\kappa_j)$, and equality $(b)$ follows by [72, Theorem 6.1, Section 6]. Here, $\{\lambda_\nu'\}_{\nu=1}^{\infty}$ represents the eigenvalues of the bilinear form $\mathcal{B}: L^2(\mathcal{W},\mathscr{B}(\mathcal{W}),\mathbb{W}) \times L^2(\mathcal{W},\mathscr{B}(\mathcal{W}),\mathbb{W}) \to \mathbb{R}$ defined as

$$
\mathcal{B}(f_1,f_2) = \frac{1}{2}E[I_2(h\left(\cdot;\mathcal{K}_{\boldsymbol{\eta}}\right))I_1(f_1)I_1(f_2)],
$$

for any $f_1, f_2 \in L^2(\mathcal{W},\mathscr{B}(\mathcal{W}),\mathbb{W})$. The terms $I_1(f_1)$ and $I_1(f_2)$ are Weiner-Itô integrals (Definition 10) defined as follows

$$
I_1(f) = \int_{\mathcal{W}} f(\boldsymbol{w})d\mathcal{Z}_{\mathbb{W}}(\boldsymbol{w}) \qquad \text{for} \qquad f \in \{f_1,f_2\}.
$$

Now, applying the stochastic integral multiplication formula [72, Section 7, Theorem 7.33], we obtain the following result:

$$
\begin{aligned}
&\frac{1}{2}E[I_2(h\left(\cdot;\mathcal{K}_{\boldsymbol{\eta}}\right))I_1(f_1)I_1(f_2)] \\
&= \frac{1}{2}\int_{\mathcal{W}^2} h\left(\boldsymbol{w}_1,\boldsymbol{w}_2;\mathcal{K}_{\boldsymbol{\eta}}\right)\left[f_1(\boldsymbol{w}_1)f_2(\boldsymbol{w}_2)+f_1(\boldsymbol{w}_2)f_2(\boldsymbol{w}_1)\right]d\mathbb{W}(\boldsymbol{w}_1)d\mathbb{W}(\boldsymbol{w}_2) \\
&= \int_{\mathcal{W}^2} h\left(\boldsymbol{w}_1,\boldsymbol{w}_2;\mathcal{K}_{\boldsymbol{\eta}}\right)f_1(\boldsymbol{w}_1)f_2(\boldsymbol{w}_2)d\mathbb{W}(\boldsymbol{w}_1)d\mathbb{W}(\boldsymbol{w}_2),
\end{aligned}
$$

where the last equation holds by the symmetry of the core function $h_2(\cdot;\mathcal{K}_{\boldsymbol{\eta}})$.

This demonstrates that the bilinear form $\mathcal{B}(f_1,f_2)$ has the same eigenvalues as those in Eqn. (20) based on $h\left(\cdot;\mathcal{K}_{\boldsymbol{\eta}}\right)$. Hence, we have that $\{\lambda_\nu\}_{\nu=1}^{\infty} = \{\lambda_\nu'\}_{\nu=1}^{\infty}$ and that

$$
\Phi(\boldsymbol{\eta}) = \Phi'(\boldsymbol{\eta}).
$$

This proves that $Z_{\mathcal{K}}$ can be represented as $G_{\mathcal{K}}$ and

$$
n \cdot \boldsymbol{U}_n^{\mathcal{K}}(W) \overset{d}{\to} G_{\mathcal{K}} = (I_2(h(\cdot;\kappa_1)), I_2(h(\cdot;\kappa_2)), ..., I_2(h(\cdot;\kappa_c)))^T.
$$

This complete the proof. $\qquad\square$

The following lemma establishes the continuity of the characteristic function.

**Lemma 11.** *Let $\Phi(\boldsymbol{\eta})$ be as defined in Eqn. (21). Then, $\Phi(\mathbf{0}) = 1$ and $\Phi(\boldsymbol{\eta})$ is continuous at $\mathbf{0} \in \mathbb{R}^c$.*

*Proof.* When $\boldsymbol{\eta} = \mathbf{0}$ with $\mathbf{0} \in \mathbb{R}^c$, the eigenvalues $\lambda_\nu = 0$ for $\nu \in \{1,2,3,...,\infty\}$. Hence, $\Phi(\mathbf{0}) = 1$ based on its definition as in Eqn. (21)

$$
\Phi(\boldsymbol{\eta}) = \prod_{\nu=1}^{\infty}\frac{\exp\left(-\iota\lambda_\nu\right)}{\sqrt{1-2\iota\lambda_\nu}} = E\left[\exp\left(\iota\cdot\sum_{\nu=1}^{\infty}\lambda_\nu(Z_\nu^2-1)\right)\right],
$$

where $\{\lambda_\nu\}_{\nu=1}^\infty$ are eigenvalues of the Hilbert-Schmidt operator $\mathcal{H}_{\mathcal{K}_{\boldsymbol{\eta}}} : L^2(\mathcal{W}, \mathscr{B}(\mathcal{W}), \mathbb{W}) \to L^2(\mathcal{W}, \mathscr{B}(\mathcal{W}), \mathbb{W})$ defined as

$$\mathcal{H}_{\mathcal{K}_{\boldsymbol{\eta}}}[f](\boldsymbol{w}_1) = \int_{-\infty}^{\infty} h(\boldsymbol{w}_1, \boldsymbol{w}_2; \mathcal{K}_{\boldsymbol{\eta}}) f(\boldsymbol{w}_2) d\mathbb{W}(\boldsymbol{w}_2),$$

where $h(\boldsymbol{w}_1, \boldsymbol{w}_2; \mathcal{K}_{\boldsymbol{\eta}}) = \sum_{j=1}^c \eta_j h(\boldsymbol{w}_1, \boldsymbol{w}_2; \kappa_j)$.

Next, we prove that $\Phi(\boldsymbol{\eta})$ is continuous at $\mathbf{0} \in \mathbb{R}^c$. It is evident that the eigenvalues $\{\lambda_\nu\}_{\nu=1}^\infty$ of the Hilbert–Schmidt operator $\mathcal{H}_{\mathcal{K}_{\boldsymbol{\eta}}}$ satisfy the summability condition [74]

$$\sum_{\nu=1}^\infty \lambda_\nu^2 < \infty.$$

Then, applying Fubini's Theorem to the term $\sum_{\nu=1}^\infty \lambda_\nu(Z_\nu^2 - 1)$, we obtain

$$E\left[\left(\sum_{\nu=1}^\infty \lambda_\nu(Z_\nu^2 - 1)\right)^2\right]$$
$$= E\left[\sum_{\nu=1}^\infty \lambda_\nu^2(Z_\nu^2 - 1)^2 + \sum_{\nu_1 \neq \nu_2}^\infty \lambda_{\nu_1}\lambda_{\nu_2}(Z_{\nu_1}^2 - 1)(Z_{\nu_2}^2 - 1)\right]$$
$$= \sum_{\nu=1}^\infty \lambda_\nu^2 E[(Z_\nu^2 - 1)^2]$$
$$= 2\sum_{\nu=1}^\infty \lambda_\nu^2.$$

Furthermore, we have, by the spectral theorem [75, Theorem 6.35, Section 6.2.1],

$$E\left[\left(\sum_{\nu=1}^\infty \lambda_\nu(Z_\nu^2 - 1)\right)^2\right] = 2\sum_{\nu=1}^\infty \lambda_\nu^2 = 2\|\mathcal{H}_{\mathcal{K}_{\boldsymbol{\eta}}}\|^2.$$

It is evident that $\lim_{\boldsymbol{\eta} \to \mathbf{0}} 2\|\mathcal{H}_{\mathcal{K}_{\boldsymbol{\eta}}}\|^2 = 0$ since $\mathcal{K}_{\boldsymbol{\eta}}(\cdot, \cdot) = \sum_{j=1}^c \eta_j \kappa_j(\cdot, \cdot)$, and thus we have $\sum_{\nu=1}^\infty \lambda_\nu(Z_\nu^2 - 1) \xrightarrow{L^2} 0$, as $\boldsymbol{\eta} \to \mathbf{0}$. Hence, by the Dominated Convergence Theorem [76, Theorem 3, Section 1.3], we have

$$\lim_{\boldsymbol{\eta} \to \mathbf{0}} \Phi(\boldsymbol{\eta}) = \lim_{\boldsymbol{\eta} \to \mathbf{0}} E\left[\exp\left(\iota \cdot \sum_{\nu=1}^\infty \lambda_\nu(Z_\nu^2 - 1)\right)\right] = 1.$$

This completes the proof that $\Phi(\cdot)$ is continuous at $\mathbf{0} \in \mathbb{R}^c$. □

## B.3   The Detailed Proofs of Lemma 3

We begin with a useful Theorem which we now present.

**Theorem 12.** *[77, Eqn. (5.7)] If the function $h(\cdot; \kappa)$ is bounded given a kernel $\kappa$, i.e., $a \leq h(\boldsymbol{w}_1, \boldsymbol{w}_2; \kappa) \leq b$, the following holds for samples $W = \{\boldsymbol{w}_1, \boldsymbol{w}_2, \ldots, \boldsymbol{w}_n\} \sim \mathbb{W}^n$,*

$$\Pr(|U_n^\kappa(W) - \theta| \geq t) \leq 2\exp\left(-2\lfloor n/2\rfloor t^2/(b-a)^2\right),$$

*where $\theta = E[h(\boldsymbol{w}_1, \boldsymbol{w}_2; \kappa)]$.*

We now present the detailed proofs of Lemma 3 as follows.

*Proof.* Denote by $\Sigma_{H_0}$ is the covariance matrix of $n\boldsymbol{U}_n^\mathcal{K}(W_{H_0})$ under null hypothesis $H_0$, consisting of (co)variances $n^2 \cdot \sigma_{H_0,a,b}$ with $1 \leq a, b \leq c$ as follows, by [78, Theorem 2, Section 1.4],

$$n^2 \cdot \sigma_{H_0,a,b} = COV\left(n\boldsymbol{U}_n^{\kappa_a}(W_{H_0}), n\boldsymbol{U}_n^{\kappa_b}(W_{H_0})\right) = n^2\binom{n}{2}^{-1}\sum_{r=1}^2 \binom{2}{r}\binom{n-2}{2-r}\zeta_r. \quad (22)$$

for some specific $\zeta_r$.

For the first-order degenerate $U$-statistics, we have $\zeta_1 = 0$ and

$$n^2 \cdot \sigma_{H_0,a,b} = n^2 \binom{n}{2}^{-1} \zeta_2 \, , \tag{23}$$

where $\zeta_2$ denotes the second-order projection term defined as

$$\zeta_2 = E_{\boldsymbol{w}_1, \boldsymbol{w}_2} \left[ h(\boldsymbol{w}_1, \boldsymbol{w}_2; \kappa_a) h(\boldsymbol{w}_1, \boldsymbol{w}_2; \kappa_b) \right] \, ,$$

with $\boldsymbol{w}_1$ and $\boldsymbol{w}_2$ independently drawn under the null hypothesis $H_0$.

A natural estimator for $\zeta_2$, based on the $U$-statistic, is given by

$$\widehat{\zeta}_2 = \binom{n}{2}^{-1} \times \sum_{1 \le i_1 < i_2 \le n} h(\boldsymbol{w}'_{i_1}, \boldsymbol{w}'_{i_2}; \kappa_a) h(\boldsymbol{w}'_{i_1}, \boldsymbol{w}'_{i_2}; \kappa_b) \, ,$$

with $\boldsymbol{w}'_{i_1}, \boldsymbol{w}'_{i_2} \in W_{H_0}$.

By Theorem 12, for any arbitrarily small positive number $\epsilon$, the term $|\widehat{\zeta}_2 - \zeta_2|$ satisfies the inequality

$$\begin{aligned}
\Pr\left( |\widehat{\zeta}_2 - \zeta_2| > \epsilon \right) \quad &\le \quad 2 \exp(-2 \lfloor n/2 \rfloor \epsilon^2 / (b^2 - a^2)^2) \\
&\to \quad 0 \quad \text{as} \quad n \to \infty \, .
\end{aligned}$$

This proves that

$$\widehat{\zeta}_2 \xrightarrow{p} \zeta_2 \quad \text{as} \quad n \to \infty \, . \tag{24}$$

For the entries $n^2 \cdot \hat{\sigma}_{H_0,a,b}$ of the estimated covariance matrix $\widehat{\Sigma}_{H_0}$, defined as

$$n^2 \cdot \hat{\sigma}_{H_0,a,b} = n^2 \binom{n}{2}^{-2} \sum_{1 \le i_1 < i_2 \le n} h(\boldsymbol{w}'_{i_1}, \boldsymbol{w}'_{i_2}; \kappa_a) h(\boldsymbol{w}'_{i_1}, \boldsymbol{w}'_{i_2}; \kappa_b) = n^2 \binom{n}{2}^{-1} \widehat{\zeta}_2 \, ,$$

we have that,

$$\hat{\sigma}^2_{H_0,a,b} \xrightarrow{p} \sigma^2_{H_0,a,b} \quad \text{as} \quad n \to \infty \, ,$$

by combining Eqns. (23) and (24).

This convergence implies that

$$\widehat{\Sigma}_{H_0} \xrightarrow{p} \Sigma_{H_0} \quad \text{as} \quad n \to \infty \, .$$

Thus, the proof is complete. $\qquad\qquad\qquad\qquad\qquad\qquad\qquad\qquad\qquad\qquad\qquad\qquad\square$

## B.4 The Detailed Proofs of Theorem 5

We begin with a useful Theorem as follows.

**Theorem 13.** *[71, Theorem 4.2.3] Under the alternative hypothesis $H_1$, for non-degenerate $h(\cdot; \kappa)$ where $E[h^2(\boldsymbol{w}_1, \boldsymbol{w}_2; \kappa)] < \infty$ for each $\kappa \in \mathcal{K}$, the following holds*

$$\sqrt{n}(\boldsymbol{U}_n^{\mathcal{K}}(W) - \boldsymbol{U}^{\mathcal{K}}(\mathbb{W})) \xrightarrow{d} \mathcal{N}(\boldsymbol{0}, \Sigma_{H_1}) \, ,$$

*where $\boldsymbol{U}^{\mathcal{K}}(\mathbb{W}) = E\left[ \boldsymbol{U}_n^{\mathcal{K}}(W) \right] = E\left[ \boldsymbol{h}(\boldsymbol{w}_1, \boldsymbol{w}_2; \mathcal{K}) \right]$ and $\Sigma_{H_1}$ is the covariance matrix of $\sqrt{n}\boldsymbol{U}_n^{\mathcal{K}}(W)$, whose entries consist of (co)variances $\sigma_{H_1,a,b}$ with $1 \le a, b \le c$ as follows*

$$\sigma_{H_1,a,b} = 4 \left( E\left[ h_1(\boldsymbol{w}_1; \kappa_a) h_1(\boldsymbol{w}_1; \kappa_b) \right] - U^{\kappa_a}(\mathbb{W}) U^{\kappa_b}(\mathbb{W}) \right) \, .$$

We now present the detailed proofs of Theorem 5 as follows.

*Proof.* The asymptotic distribution of $\boldsymbol{U}_n^{\mathcal{K}}(W)$ follows directly from Theorem 13. In the following, we analyze the asymptotic distribution of $T_n^{\mathcal{K}}(W)$.

By applying the Taylor series expansion of $T_n^{\mathcal{K}}(W)$ at $U^{\mathcal{K}}(\mathbb{W})$, we obtain

$$
\begin{aligned}
T_n^{\mathcal{K}}(W) &= n^2 (U^{\mathcal{K}}(\mathbb{W}))^T \widehat{\Sigma}_{H_0}^{-1} U^{\mathcal{K}}(\mathbb{W}) + 2n^2 \left(U_n^{\mathcal{K}}(W) - U^{\mathcal{K}}(\mathbb{W})\right)^T \widehat{\Sigma}_{H_0}^{-1} U^{\mathcal{K}}(\mathbb{W}) \\
&\quad + n^2 \left(U_n^{\mathcal{K}}(W) - U^{\mathcal{K}}(\mathbb{W})\right)^T \widehat{\Sigma}_{H_0}^{-1} \left(U_n^{\mathcal{K}}(W) - U^{\mathcal{K}}(\mathbb{W})\right) .
\end{aligned}
$$

Furthermore, it is evident that $\|U_n^{\mathcal{K}}(W) - U^{\mathcal{K}}(\mathbb{W})\|_2 = O_p(1/\sqrt{n})$ from Theorem 13. Building on this, we have

$$
\begin{aligned}
&n^{-3/2} \left(T_n^{\mathcal{K}}(W) - n^2 \left(U^{\mathcal{K}}(\mathbb{W})\right) \widehat{\Sigma}_{H_0}^{-1} U^{\mathcal{K}}(\mathbb{W})\right) \\
&= 2n^{1/2} \left(U_n^{\mathcal{K}}(W) - U^{\mathcal{K}}(\mathbb{W})\right)^T \widehat{\Sigma}_{H_0}^{-1} U^{\mathcal{K}}(\mathbb{W}) \\
&\quad + n^{1/2} \left(U_n^{\mathcal{K}}(W) - U^{\mathcal{K}}(\mathbb{W})\right)^T \widehat{\Sigma}_{H_0}^{-1} \left(U_n^{\mathcal{K}}(W) - U^{\mathcal{K}}(\mathbb{W})\right) \\
&= 2n^{1/2} \left(U_n^{\mathcal{K}}(W) - U^{\mathcal{K}}(\mathbb{W})\right)^T \widehat{\Sigma}_{H_0}^{-1} U^{\mathcal{K}}(\mathbb{W}) + O(1/\sqrt{n}) .
\end{aligned}
$$

By applying Lemma 3 and Theorem 13, and invoking Slutsky's theorem [65], we obtain the following asymptotic distribution

$$
n^{-3/2} \left(T_n^{\mathcal{K}}(W) - n^2 \left(U^{\mathcal{K}}(\mathbb{W})\right) \widehat{\Sigma}_{H_0}^{-1} U^{\mathcal{K}}(\mathbb{W})\right) \xrightarrow{d} \mathcal{N}(0, \sigma_{H_1}^2) ,
$$

where $\sigma_{H_1}^2 = 4 \left(U^{\mathcal{K}}(\mathbb{W})\right)^T \Sigma_{H_0}^{-1} \Sigma_{H_1} \Sigma_{H_0}^{-1} U^{\mathcal{K}}(\mathbb{W})$. $\qquad\square$

## B.5 The Detailed Proofs of Theorem 6

The proof of Theorem 6 builds upon the results established in [7, Proposition 1]. We present the detailed proofs of Theorem 6 as follows.

*Proof.* Under the null hypothesis $H_0$, and following the analysis in [7, Appendix F.1] for MMD and HSIC, we have that the corresponding test $U$-statistic $nU_n^{\kappa}(W_{te})$ shares the same asymptotic distribution as its $b$-th wild bootstrap $U$-statistic $nU_n^{\kappa,b}(W_{te})$ defined as

$$
n \cdot U_n^{\kappa,b}(W_{te}) = n \cdot \binom{n}{2}^{-1} \sum_{1 \le i_1 < i_2 \le n} \epsilon_{i_1} \epsilon_{i_2} h(\boldsymbol{w}_{i_1}, \boldsymbol{w}_{i_2}; \kappa) ,
$$

for all $b \in [B]$ and all $\kappa \in \mathcal{K}$. Consequently, by defining the multiple kernel forms

$$
\begin{aligned}
n \cdot U_n^{\mathcal{K}}(W_{te}) &= \left(n \cdot U_n^{\kappa_1}(W_{te}), n \cdot U_n^{\kappa_2}(W_{te}), ..., n \cdot U_n^{\kappa_c}(W_{te})\right)^T , \\
n \cdot U_n^{\mathcal{K},b}(W_{te}) &= \left(n \cdot U_n^{\kappa_1,b}(W_{te}), n \cdot U_n^{\kappa_2,b}(W_{te}), ..., n \cdot U_n^{\kappa_c,b}(W_{te})\right)^T ,
\end{aligned}
$$

it follows that the multivariate $U$-statistics $nU_n^{\mathcal{K}}(W_{te})$ and $nU_n^{\mathcal{K},b}(W_{te})$ share the same asymptotic distribution by Cramér-Wold Theorem, as shown in [31, Section 6.1] using the fact that the U-statistics are linear with respect to their kernel parameter. Furthermore, in the asymptotic manner, we have $\widehat{L}_{H_0} \to L_{H_0}$ established in Lemma 3 with $\Sigma_{H_0} = L_{H_0} L_{H_0}$ and $\widehat{\Sigma}_{H_0} = \widehat{L}_{H_0} \widehat{L}_{H_0}$. Based on Slutsky's Theorem [65], it follows directly from Theorem 2 that

$$
\begin{aligned}
n \cdot \widehat{L}_{H_0} U_n^{\mathcal{K}}(W_{te}) &\xrightarrow{d} L_{H_0}^{-1} G_{\mathcal{K}} , \\
n \cdot \widehat{L}_{H_0} U_n^{\mathcal{K},b}(W_{te}) &\xrightarrow{d} L_{H_0}^{-1} G_{\mathcal{K}} .
\end{aligned}
$$

According to the definition in Eqn. (9), the test statistic and the wild bootstrap statistic can be expressed as follows

$$
\begin{aligned}
\mathcal{T} &= \| \max(n \cdot \mathrm{diag}(\boldsymbol{F}_{tr}) \widehat{L}_{H_0}^{-1} U_n^{\mathcal{K}}(W_{te}), \boldsymbol{0})\|_2^2 , \\
\mathcal{T}^b &= \left\| \max\left(n \cdot \mathrm{diag}(\boldsymbol{F}_{tr}) \widehat{L}_{H_0}^{-1} U_n^{\mathcal{K},b}(W_{te}), \boldsymbol{0}\right) \right\|_2^2 .
\end{aligned}
$$

It is observed that the mapping $x \mapsto \max(x, 0)$ is continuous. Consequently, $\mathcal{T}$ and $\mathcal{T}^b$ are continuous functions of $n \cdot \widehat{L}_{H_0}^{-1} U_n^{\mathcal{K}}(W_{te})$ and $n \cdot \widehat{L}_{H_0}^{-1} U_n^{\mathcal{K},b}(W_{te})$, respectively.

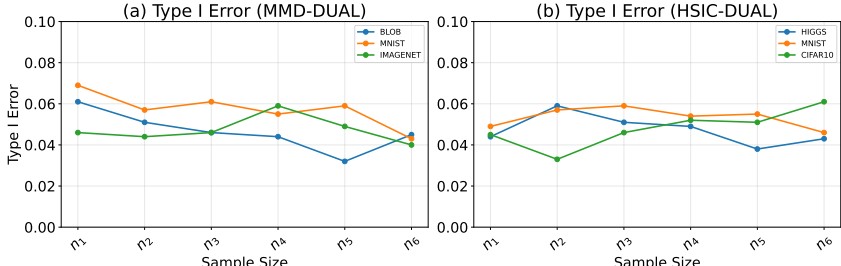

**Figure 5:** Two-sample testing: ($a$) Type I error checking experiments on dataset BLOB, MNIST and ImageNet; and Independence testing: ($b$) Type I error checking experiments on dataset Higgs, MNIST and CIFAR10. The Type I error results are averaged over 1,000 repetitions under the significant level $\alpha = 0.05$. $n_1 - n_6$ refers to a set of six sample sizes associated with each dataset presented in Figure 3, where the specific sample sizes vary across datasets.

Finally, by *Continuous Mapping Theorem* [68], we have

$$\mathcal{T} \overset{d}{\to} \left\| \max \left( \mathrm{diag}(\boldsymbol{F}_{tr}) \boldsymbol{L}_{H_0}^{-1} G_{\mathcal{K}}, \boldsymbol{0} \right) \right\|_2^2 \;,$$

$$\mathcal{T}^b \overset{d}{\to} \left\| \max \left( \mathrm{diag}(\boldsymbol{F}_{tr}) \boldsymbol{L}_{H_0}^{-1} G_{\mathcal{K}}, \boldsymbol{0} \right) \right\|_2^2 \;.$$

This completes the proof. □

## C  Supplementary Experimental Results

In this Section, we will illustrate the details on the experiments we conducted on two-sample testing and independence testing. Moreover, we present the results of extra Type-I error check and ablation experiments on additional datasets.

### C.1  Experimental Details

**Initialization of Kernels.** We allow 6 different categories of kernels to be aggregated together: O+L, O+G, O+M, R+L, R+G and R+M, where O denotes original features, R denotes representation of original features learned by deep neural network [6], L denotes Laplacian kernel, G denotes Gaussian kernel and M denotes Mahalanobis kernel. For the initialization of bandwidths for each categories, we can either select from scaled median heuristics [7, 9] or by grid-search [51]. For the median heuristics-based approach, specifically, we compute the 0.05 and 0.95 quantiles of the pairwise distances between samples, and then select bandwidths uniformly between the minimum of the 0.05 quantile and the maximum of the 0.95 quantile. With a fixed kernel set size $c$, this procedure yields the same kernel set for the same data, ensuring both performance and reproducibility.

In Table 2, we further analyze the sensitivity of the kernel set initialization. Specifically, we compare the test powers of the MMD-DUAL, MMDAgg, and MMD-FUSE methods when the kernel sets (with 10 kernels) are initialized using either the heuristic approach or random initialization. From the result, we find that across all methods, heuristic initialization yields higher and more stable test power, as indicated by lower standard deviations, compared to random initialization. Moreover, randomly initializing the kernel sets leads to a noticeable drop in performance for all methods, generally resulting in lower test power and higher standard deviations. However, MMD-DUAL differs from MMDAgg and MMD-FUSE in that it performs kernel optimization. While MMDAgg and MMD-FUSE fully rely on the initial kernel sets, MMD-DUAL can adaptively optimize over a diverse kernel pool even when the initialization is not ideal. This enables MMD-DUAL to maintain relatively high and stable test power under random initialization.

**Learnable parameters for kernel learning.** As we use four different kinds of kernels, we will list the parameters for each kernel below:

1. Gaussian kernel: $\kappa_i(x, y) = \exp(-||x - y||^2 / \sigma_i^2)$, where bandwidth $\sigma_i$ is the trainable parameters.

**Table 2:** Test power±standard deviation for dataset BLOB under heuristic kernel initialization and random kernel initialization. $N$ is the number of pairs of two samples.

| Heuristic | N=50 | N=100 | N=150 | N=200 | N=250 | N=300 |
|---|---|---|---|---|---|---|
| MMD-DUAL | $0.134 \pm 0.016$ | $\mathbf{0.454 \pm 0.022}$ | $\mathbf{0.750 \pm 0.028}$ | $\mathbf{0.915 \pm 0.016}$ | $\mathbf{0.983 \pm 0.005}$ | $\mathbf{0.998 \pm 0.001}$ |
| MMDAgg | $0.124 \pm 0.055$ | $0.270 \pm 0.076$ | $0.539 \pm 0.073$ | $0.764 \pm 0.053$ | $0.941 \pm 0.026$ | $0.987 \pm 0.008$ |
| MMD-FUSE | $\mathbf{0.153 \pm 0.055}$ | $0.346 \pm 0.065$ | $0.673 \pm 0.084$ | $0.863 \pm 0.042$ | $0.972 \pm 0.011$ | $\mathbf{0.998 \pm 0.002}$ |
| **Random** | | | | | | |
| MMD-DUAL | $0.084 \pm 0.067$ | $\mathbf{0.365 \pm 0.123}$ | $\mathbf{0.420 \pm 0.084}$ | $\mathbf{0.585 \pm 0.038}$ | $\mathbf{0.737 \pm 0.055}$ | $\mathbf{0.825 \pm 0.061}$ |
| MMDAgg | $0.058 \pm 0.028$ | $0.148 \pm 0.054$ | $0.210 \pm 0.056$ | $0.410 \pm 0.163$ | $0.675 \pm 0.158$ | $0.733 \pm 0.169$ |
| MMD-FUSE | $\mathbf{0.095 \pm 0.067}$ | $0.098 \pm 0.048$ | $0.203 \pm 0.152$ | $0.439 \pm 0.181$ | $0.656 \pm 0.163$ | $0.740 \pm 0.083$ |

2. Laplacian kernel: $\kappa_i(x,y) = \exp(-||x-y||/\sigma_i)$, where bandwidth $\sigma_i$ is the trainable parameters.

3. Mahalanobis kernel: $\kappa_i(x,y) = \exp(-(x-y)^T \Sigma_i^{-1}(x-y))$, where the covariance matrix $\Sigma_i$ is the trainable parameters.

4. Deep kernel: $\kappa_i(x,y) = [(1-\epsilon)\exp(-||\phi_{\omega_i}(x) - \phi_{\omega_i}(y)||^2/\sigma_{\phi_i}^2) + \epsilon]\exp(-||x-y||^2/\sigma_i^2)$, where the parameters $\omega_i$ in the deep neural network $\phi_{\omega_i}$, bandwidths $\sigma_{\phi_i}$ and $\sigma_i$, and the weight $\epsilon$ are all trainable.

In the reproducible code repository (it can be redirected from the provided code link), we use all four kernels. However, for simplicity to use, even gaussian kernels with different bandwidths can achieve high performance.

**Two-sample Testing Baselines and Experiments.** In two-sample testing experiments, we compare our proposed MMD-DUAL with 4 state-of-the-art baselines. The implementations of AutoTST [63], MMDAgg [8], MMD-D [6] and MMD-FUSE [9] can all be found in the GitHub link displayed in their papers' Abstract. For MMDAgg and MMD-FUSE, we use the default settings of total number of twenty bandwidths (ten O+L and ten O+G). For the implementation of our proposed MMD-DUAL, we use one O+G and one O+M for BLOB dataset, use each of the O+L, O+G, O+M, R+L, R+G and R+M for MNIST dataset and use one O+G, one O+M and one R+G for ImageNet dataset. For the representation model architecture, we follow the implementation of [6] in BLOB dataset and [12] in image dataset. The learning rate is $5e^{-4}$ for BLOB and $5e^{-5}$ for MNIST and ImageNet. For all the two-sample testing experiments, we conduct each experiment with ten different seeds, and for each seed, we perform the testing data selection and two-sample testing process for 100 times. In total, the results are all averaging over 1,000 repetitions.

**Independence Testing Baselines and Experiments.** In independence testing experiments, we compare our proposed HSIC-DUAL with 3 state-of-the-art baselines. The implementations of FSIC [13], HSICAgg [7] and HSIC-O [15] can all be found in the GitHub link displayed in their papers' Abstract. For HSIC-Agg, we use the default settings of total number of nine kernel O+G. For the implementation of our proposed HSIC-DUAL, we use four O+G, four O+L, five O+M for Higgs dataset, use two O+G, two O+L, three R+G and three R+L for MNIST dataset and use four O+G, four O+L, four R+G and four R+L for CIFAR10 dataset. The representation model architecture of original features are exactly same as the two-sample testing settings. The implementation of [6] in Higgs dataset and [12] in image dataset. In independence testing, we only apply representation encoder model on the original features, not on labels. The learning rate is $5e^{-4}$ for all three datasets. Moreover, for all the independence testing experiments, we conduct each experiment with ten different seeds, and for each seed, we perform the testing data selection and independence testing process for 100 times. In total, the results are all averaging over 1,000 repetitions.

**Two-sample and Independence Testing Datasets.** All the sample sizes $n$ in the Figure 3 represents the number of samples we use in both the training phase and testing phase. Thus, for baselines without data-splitting (e.g., MMDAgg, HSICAgg, FSIC and MMD-FUSE), we use $2n$ samples to ensure the fairness that there are same total samples included in the whole testing experiments. For two-sample testing, we use the BLOB dataset generated in the same way as [3, 6], use the adversarial MNIST the same as [6, 8] and use the ImageNetV2 the same as [6]. For independence testing, we generate different linear perturbations on each dimensions of the original Higgs data to measure the test power [79]. For image datasets, we generate the corruption of labels with percentage of 75% (only 25% of the images have their original labels) to measure the test power, followed by [7]. For

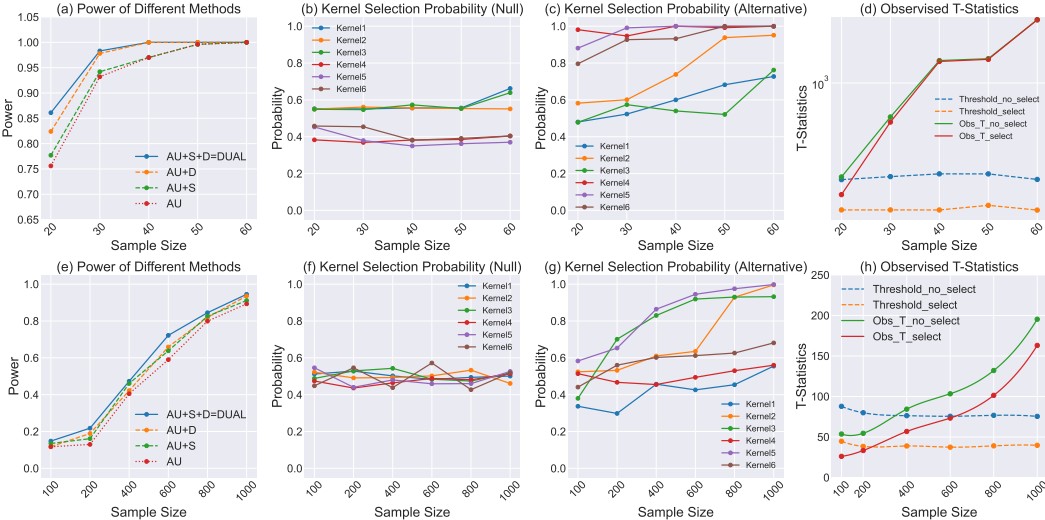

**Figure 6:** Ablation Study on the effectiveness of diversity and selection inference on two more image datasets. $(a-d)$ are ablation study for MMD-DUAL on MNIST; $(e-h)$ are ablation study for HSIC-DUAL on MNIST. $(a,e)$ Test power for model variants: AU represents simple Aggregated $U$-Statistics; S represents selection inference technique; D represents considering diversity into AU; AU+S+D refers to our proposed DUAL.

all three datasets under the null hypothesis in two-sample testing, we draw two samples from same distribution (e.g., both from adversarial images). For all three datasets under the null hypothesis in the independence testing, we independently and separately draw two samples from the original datasets.

**Implementation resources.** The experiments of the work are conducted on two platforms. One platform is an Nvidia-4090 GPU PC with Pytorch framework. Another platform is a High-performance Computer cluster with several Nvidia-A100 GPUs with Pytorch framework. The memory of two platforms are both 64 GB. The storage of disk of two platforms are both over 4 TB.

## C.2 Additional Experiments Results

**Type-I error Check.** In Figure 5, we conduct the Type-I error check under the null hypothesis, the Type-I error of DUAL is controlled at the desired significance level $\alpha = 0.05$. The choice of significance level value and the procedure of Type-I error check also follow all the previous works in two-sample testing and independence testing [1, 3, 7, 8, 6].

**Additional Ablation Results.** In Figure 6, we conduct additional ablation experiments to show that, the analysis in Section 6.2 is consistent across different datasets and different testing methods, proving the effectiveness of the learned diverse kernels and selection inference technique from DUAL.

**Time Complexity Results.** In Table 3, we present the running time of MMD-DUAL, MMDAgg and MMD-FUSE across different datasets and sample sizes. In this experiment, all three methods are initialized with the same kernel set, following the experimental setup of [8]. From the result, we observe that all methods have similar testing times. The key difference lies in the training phase: MMDAgg and MMD-FUSE do not involve any optimization and therefore incur no training time, while MMD-DUAL includes a kernel optimization phase, resulting in a training cost.

## D Analytical Support for Ablation Study Results

To facilitate comprehension and offer deeper insight, we present a simple example illustrating the advantages of our testing approach, which incorporates the alignment vector with wild bootstrap. This example serves as a more detailed analytical explanation and provides additional support for the ablation study results shown in Figures 4 and 6.

**Example 1.** *For simplicity, we assume that each component in the vector* $n\widehat{\boldsymbol{L}}_{H_0}^{-1}\boldsymbol{U}_n^{\mathcal{K}}(W_{te})$ *are mutually independent and share the same scale. For each component* $\left(n\widehat{\boldsymbol{L}}_{H_0}^{-1}\boldsymbol{U}_n^{\mathcal{K}}(W_{te})\right)_i$ *with*

**Table 3:** Running Time (seconds) per two-sample test trial. We record the total training time and testing time for three methods across 100 trials, then we divide each total time by 100. For MMDAgg and MMD-FUSE, there are no training time, and the number of samples used in testing will be double than that used in MMD-DUAL testing, since MMD-DUAL has a half-half train-test splitting process.

| Running Time (s) | BLOB | | | | | | MNIST | | | | | | ImageNet | | | | | |
|---|---|---|---|---|---|---|---|---|---|---|---|---|---|---|---|---|---|---|
| | N=50 | N=100 | N=150 | N=200 | N=250 | N=300 | N=20 | N=30 | N=40 | N=50 | N=60 | N=70 | N=400 | N=500 | N=600 | N=700 | N=800 | N=900 |
| **Training** | | | | | | | | | | | | | | | | | | |
| MMD-DUAL | 0.047 | 0.048 | 0.048 | 0.050 | 0.052 | 0.054 | 1.291 | 1.324 | 1.340 | 1.357 | 1.360 | 1.377 | 9.682 | 11.081 | 13.546 | 15.996 | 18.727 | 21.983 |
| MMDAgg | 0 | 0 | 0 | 0 | 0 | 0 | 0 | 0 | 0 | 0 | 0 | 0 | 0 | 0 | 0 | 0 | 0 | 0 |
| MMD-FUSE | 0 | 0 | 0 | 0 | 0 | 0 | 0 | 0 | 0 | 0 | 0 | 0 | 0 | 0 | 0 | 0 | 0 | 0 |
| **Testing** | | | | | | | | | | | | | | | | | | |
| MMD-DUAL | 0.004 | 0.004 | 0.004 | 0.004 | 0.004 | 0.004 | 0.018 | 0.018 | 0.018 | 0.018 | 0.018 | 0.018 | 0.707 | 0.807 | 1.024 | 1.110 | 1.274 | 1.471 |
| MMDAgg | 0.004 | 0.004 | 0.004 | 0.004 | 0.004 | 0.004 | 0.029 | 0.031 | 0.031 | 0.031 | 0.031 | 0.032 | 0.763 | 0.957 | 1.167 | 1.333 | 1.448 | 1.594 |
| MMD-FUSE | 0.001 | 0.002 | 0.002 | 0.002 | 0.003 | 0.004 | 0.028 | 0.028 | 0.028 | 0.028 | 0.028 | 0.028 | 0.680 | 0.843 | 0.985 | 1.172 | 1.406 | 1.552 |

$i \in [c]$, *we consider its wild bootstrap value* $\left(n\widehat{\boldsymbol{L}}_{H_0}^{-1}\boldsymbol{U}_n^{\mathcal{K},b}(W_{te})\right)_i$ *in b-th iteration with* $b \in [B]$, *it follows that*

$$\Pr\left(\left(n\widehat{\boldsymbol{L}}_{H_0}^{-1}\boldsymbol{U}_n^{\mathcal{K},b}(W_{te})\right)_i \geq 0\right) = \Pr\left(\left(n\widehat{\boldsymbol{L}}_{H_0}^{-1}\boldsymbol{U}_n^{\mathcal{K},b}(W_{te})\right)_i < 0\right) = 0.5,$$

*based on the i.i.d. Rademacher random variables (which are symmetric about 0). Given the fixed vector* $\boldsymbol{F}_{tr} \in \{-1, +1\}^c$ *from training phase, the probability that the sign of the i-th component of* $n\widehat{\boldsymbol{L}}_{H_0}^{-1}\boldsymbol{U}_n^{\mathcal{K},b}(W_{te})$ *matches the corresponding entry in* $\boldsymbol{F}_{tr}$ *is thus* $0.5$, *i.e.,*

$$\Pr\left(sgn\left(n\widehat{\boldsymbol{L}}_{H_0}^{-1}\boldsymbol{U}_n^{\mathcal{K},b}(W_{te})\right)_i = F_{tr,i}\right) = 0.5,$$

*where* $sgn\left(n\widehat{\boldsymbol{L}}_{H_0}^{-1}\boldsymbol{U}_n^{\mathcal{K},b}(W_{te})\right)$ *is denoted by* $\boldsymbol{F}_{te}^b$ *in the wild bootstrap process. As a result, the probability that* $F_i^b = \mathbb{I}[F_{te,i}^b = F_{tr,i}] = 0$ *is equal to* $0.5$. *We denote* $\tau_\alpha$ *as the asymptotic value of* $\hat{\tau}_\alpha$ *with* $B = \infty$ *in Eqn.* (10) *and denote by* $\tau_\alpha^{\mathrm{NA}}$ *the asymptotic threshold of statistic* $n^2\|\widehat{\boldsymbol{L}}_{H_0}^{-1}\boldsymbol{U}_n^{\mathcal{K}}(W_{te})\|_2^2$, *i.e., the statistic without selection inference. Asymptotically, as* $c \to \infty$, *it follows that*

$$\tau_\alpha = 0.5\tau_\alpha^{\mathrm{NA}}.$$

*In contrast, under the alternative hypothesis* $H_1$, *the asymptotic distributions of the statistic vectors* $n\widehat{\boldsymbol{L}}_{H_0}^{-1}\boldsymbol{U}_n^{\mathcal{K}}(W_{te})$ *and* $n\widehat{\boldsymbol{L}}_{H_0}^{-1}\boldsymbol{U}_n^{\mathcal{K}}(W_{tr})$ *deviate from* $\boldsymbol{0}$, *as can be inferred from Theorem* 5. *Denote by* $\boldsymbol{F}_{te} = \mathrm{sgn}\left(\widehat{\boldsymbol{L}}_{H_0}^{-1}\boldsymbol{U}_n^{\mathcal{K}}(W_{te})\right)$ *and* $\boldsymbol{F}_{tr} = \mathrm{sgn}\left(\widehat{\boldsymbol{L}}_{H_0}^{-1}\boldsymbol{U}_n^{\mathcal{K}}(W_{tr})\right)$ *the sign vector. As a consequence, the probability that the signs of the i-th components of two vectors disagree, i.e.,* $\Pr(F_i = \mathbb{I}[F_{te,i} = F_{tr,i}] = 0)$, *is reduced to some* $\beta < 0.5$. *Asymptotically, as* $c \to \infty$, *it follows that*

$$\mathcal{T} = \|n \cdot \boldsymbol{F} \circ \widehat{\boldsymbol{L}}_{H_0}^{-1}\boldsymbol{U}_n^{\mathcal{K}}(W_{te})\|_2^2 = (1-\beta)n^2\|\widehat{\boldsymbol{L}}_{H_0}^{-1}\boldsymbol{U}_n^{\mathcal{K}}(W_{te})\|_2^2,$$

*which results in an improvement in the test power as follows*

$$
\begin{aligned}
\Pr(\mathcal{T} > \tau_\alpha) - \Pr\left(n^2\|\widehat{\boldsymbol{L}}_{H_0}^{-1}\boldsymbol{U}_n^{\mathcal{K}}(W_{te})\|_2^2 > \tau_\alpha^{\mathrm{NA}}\right) &= Pr(\mathcal{T} > \tau_\alpha) - \Pr\left(\frac{\mathcal{T}}{1-\beta} > 2\tau_\alpha\right) \\
&= Pr(\mathcal{T} > \tau_\alpha) - \Pr\left(\mathcal{T} > 2(1-\beta)\tau_\alpha\right) \\
&> 0.
\end{aligned}
$$

In this example, we assume mutual independence to derive $\Pr\left(F_i^b = \mathbb{I}[F_{te,i}^b = F_{tr,i}] = 0\right) = 0.5$, a condition that is challenging to achieve but is empirically observed in practice due to the de-correlation property of $\widehat{\boldsymbol{L}}_{H_0}^{-1}$ in practice, as illustrated in Figures 4 and 6 (second column). Furthermore, we investigate the probability $\beta$, which is larger than 0.5 and increases as sample size increases as shown in Figures 4 and 6 (third column).

## E  Potential Applications of the Proposed Multivariate U-Statistics

As noted in prior work, MMD-GAN [80] is a generative model that relies on the standard MMD loss. In principle, this loss could be replaced with our proposed multivariate $U$-statistic in (4), i.e., $T_n^K(W)$. This statistic aggregates information from MMD statistics computed with multiple kernels,

allowing it to capture richer information about the discrepancy between the target and generative distributions. By minimizing $T_n^K(W)$, the generator may produce samples that better align with the target distribution compared to using the standard MMD alone.

Moreover, the proposed MMD-DUAL and HSIC-DUAL could potentially be applied to various domains that require statistical two-sample or independence testing methods. For example, *MMD-DUAL* (two-sample) could be used for tasks involving the comparison of a trusted reference against incoming data, such as out-of-distribution detection [81], adversarial image detection [82], and distribution alignment in transfer learning [83]. Similarly, *HSIC-DUAL* (independence) could be useful in domains that require mitigating or verifying statistical dependence, including domain generalization [84], causal discovery [85], machine unlearning [86], and trustworthy machine learning [87].

