# OpenReview forum: "DUAL: Learning Diverse Kernels for Aggregated Two-sample and Independence Testing"
_NeurIPS.cc/2025/Conference — NeurIPS 2025 poster_

### Official Review · Reviewer_BHAJ · 2025-06-27

**Clarity:** 3
**Significance:** 3
**Originality:** 3
**Rating:** 5
**Confidence:** 4

**Summary:**

This paper addresses the challenge of improving kernel-based nonparametric hypothesis testing, specifically for two-sample and independence tests, by introducing a method called Diverse U-statistic Aggregation with Learned Kernels (DUAL). TThe proposed method leverages a data-splitting approach to learn diverse kernels during training and employs selection inference during testing to focus on the most informative kernels. Theoretical guarantees for Type-I error control and asymptotic power are provided, and extensive experiments demonstrate superior performance over state-of-the-art methods across multiple benchmarks.

**Questions:**

1. How does the method scale with larger kernel pools (e.g., c > 20 ) or high-dimensional data? Could the authors provide guidelines for choosing c in practice?

2. The paper mentions gradient-based kernel learning but does not discuss sensitivity to initialization or optimization challenges.

3. The experiments include MMD-D (deep kernels) as a baseline. How does DUAL compare when combined with deep kernels? Is there a synergy?

**Ethical Concerns:**

["NO or VERY MINOR ethics concerns only"]

**Final Justification:**

The author has addressed most of my concerns, and I am inclined to maintain my positive score.

**Limitations:**

The authors acknowledge limitations (e.g., redundancy in kernels) but could better discuss computational trade-offs and assumption violations.

**Quality:**

4

**Strengths And Weaknesses:**

Strengths：

1.The paper is technically sound, with rigorous theoretical proofs and comprehensive experiments.

2.The paper is well-structured, with clear motivations, methodology, and results.

3.The work addresses a practical limitation in kernel aggregation methods, offering a solution that enhances the power of widely used tests like MMD and HSIC.

Weaknesses：

1.While the method is computationally feasible, the impact of kernel pool size c on scalability for very large datasets or high-dimensional kernels could be further explored.

2.The paper does not deeply discuss how sensitive the method is to the choice of kernel initialization or the number of kernels c. This could affect reproducibility in practice.

---

> ### Author Rebuttal · Authors · 2025-07-31
>
> Thank you very much for your thoughtful and constructive review. We truly appreciate your time and effort. Below, we answer your questions in detail.
>
> **Weakness 1 + Question 1:**
> > While the method is computationally feasible, the impact of kernel pool size c on scalability for very large datasets or high-dimensional kernels could be further explored.
>
> > How does the method scale with larger kernel pools (e.g., $c$ > 20 ) or high-dimensional data? Could the authors provide guidelines for choosing $c$ in practice?
>
> **Response 1 + 1:**
>
> Thank you for the question regarding the scalability of our method. It is true that using a very large candidate kernel pool may lead to computational inefficiency. However, in the context of two-sample and independence testing, the kernel sets are selected using heuristics, following the methodology of [1,2,3], and increasing the number of kernels beyond a moderate size does not lead to noticeable gains in test power. As shown in the empirical results from [1], specifically Figure 6 in Section 5.7, increasing the number of kernels from 10 to 100 and even to 1000 does not improve power. **This supports the choice of using a small number of kernels (e.g., $c = 10$)**, as there is nothing to gain empirically from using a finer discretization for the bandwidths of kernels. In fact, the referenced study even considers aggregating 12,000 kernels.
>
> For high-dimensional data, kernel methods scale linearly with the input dimension, but this cost only arises during the computation of the kernel matrix (often via pairwise distances), which can be implemented very efficiently and is rarely the bottleneck in practice.
>
> We will add a discussion in the next version on these points to illustrate the scalability of our method.
>
> **Weakness 2 + Question 2:**
> > The paper does not deeply discuss how sensitive the method is to the choice of kernel initialization or the number of kernels $c$. This could affect reproducibility in practice.
>
> > The paper mentions gradient-based kernel learning but does not discuss sensitivity to initialization or optimization challenges.
>
> **Response 2 + 2:**
>
> Thanks for the question on the initialization and number of kernel sets. As discussed above, in practice, using a small set of kernels (e.g., 10 kernels) is enough [1]. **For initializing the kernel set, we follow the methodology of [1,2,3] and use a heuristic-based approach.** Specifically, we compute the 0.05 and 0.95 quantiles of the pairwise distances between samples, and then select bandwidths uniformly between the minimum of the 0.05 quantile and the maximum of the 0.95 quantile. With a fixed kernel set size $c$, this procedure produces the same kernel set for the same data, ensuring both performance and reproducibility.
>
> In the following table, we further analyze the sensitivity of the kernel set initialization. Specifically, we compare the test powers of the DUAL, Agg, and FUSE methods when the kernel sets (with 10 kernels) are initialized using either the **heuristic approach or random initialization:**
> | Test power | BLOB | | | | | |
> |------------|------|-------|-------|-------|-------|-------|
> |**Heuristic**| N=50 | N=100 | N=150 | N=200 | N=250 | N=300 |
> | MMD-DUAL | 0.134±0.016 | **0.454±0.022** | **0.750±0.028** | **0.915±0.016** | **0.983±0.005** | **0.998±0.001** |
> | MMDAgg | 0.124±0.055 | 0.270±0.076 | 0.539±0.073 | 0.764±0.053 | 0.941±0.026 | 0.987±0.008 |
> | MMD-FUSE | **0.153±0.055** | 0.346±0.065 | 0.673±0.084 | 0.863±0.042 | 0.972±0.011  | **0.998±0.002** |
> |**Random**|  |  |  |  |  |  |
> | MMD-DUAL | 0.084±0.067 | **0.365±0.123** | **0.420±0.084** | **0.585±0.038** | **0.737±0.055** | **0.825±0.061** |
> | MMDAgg | 0.058±0.028 | 0.148±0.054 | 0.210±0.056 | 0.410±0.163 | 0.675±0.158 | 0.733±0.169 |
> | MMD-FUSE | **0.095±0.067** | 0.098±0.048 | 0.203±0.152 | 0.439±0.181 | 0.656±0.163 | 0.740±0.083 |
>
> From the table, we observe the following:
> - **Heuristic initialization:** Across all methods, heuristic initialization yields higher and more stable test power, as indicated by lower standard deviations, compared to random initialization.
> - **Random initialization:** Randomly initializing the kernel sets leads to a noticeable drop in performance for all methods, generally resulting in **lower test power and higher standard deviations**. However, MMD-DUAL differs from MMDAgg and MMD-FUSE in that it performs kernel optimization. **While MMDAgg and MMD-FUSE fully rely on the initial kernel sets, MMD-DUAL can adaptively optimize over a diverse kernel pool even when the initialization is not ideal**. This enables MMD-DUAL to maintain relatively high and stable test power under random initialization.
>
> **We will emphasize that we follow the methodology of [1,2,3] in adopting a heuristic-based initialization approach, and we will include these experimental details and dicussions in the updated version of the paper.**
>
> **Question 3:**
> > The experiments include MMD-D (deep kernels) as a baseline. How does DUAL compare when combined with deep kernels? Is there a synergy?
>
> **Response 3:**
>
> Thank you for the question regarding the combination of deep kernels. We do aggregate over deep kernels in our method. In our experiments, we can use four categories of kernels: Gaussian, Laplace, Mahalanobis, and deep kernels. We will include these important implementation details in the next revision.
>
> **Question 4:**
> >The authors acknowledge limitations (e.g., redundancy in kernels) but could better discuss computational trade-offs and assumption violations.
>
> **Response 4:**
>
> Thank you for the suggestion. We will include a discussion on the computational trade-offs as noted in Response 1+1, and also discuss the assumptions made in our paper in more detail. The boundedness assumption on the kernel is standard [1–6]. The kernels used in our paper, including Gaussian, Laplace, Mahalanobis, and deep kernels, are all bounded. For unbounded kernels, such as $\kappa(x,y)=\\|x - y\\|^2_2$, the data can be normalized to a fixed range to ensure this assumption is satisfied. Assumption 1 is used to analyze the consistency of our test as $n \to \infty$. As shown in Figures 4(c) and 4(g), this assumption is easily satisfied in practice, since the probability of selecting at least one kernel quickly approaches 1 even with a limited sample size.
>
> ---
>
> [1]. A. Schrab, I. Kim, M. Albert, B. Laurent, B. Guedj, and A. Gretton. MMD aggregated two-sample test. Journal of Machine Learning Research, 2023.
>
> [2]. F. Biggs, A. Schrab, and A. Gretton. MMD-Fuse: Learning and combining kernels for two sample testing without data splitting. In NeurIPS, 2023.
>
> [3]. A. Schrab, I. Kim, B. Guedj, and A. Gretton. Efficient aggregated kernel tests using incomplete U-statistics. In NeurIPS 2022.
>
> [4]. A. Gretton, K. M. Borgwardt, M. J. Rasch, B. Schölkopf, and A. J. Smola. A kernel method for the two-sample-problem. In NeurIPS, 2006.
>
> [5]. A. Gretton, K. Fukumizu, C. H. Teo, L. Song, B. Schölkopf, and A. J. Smola. A kernel statistical test of independence. In NeurIPS, 2007.
>
> [6]. F. Liu, W. Xu, J. Lu, G. Zhang, A. Gretton, and D. J. Sutherland. Learning deep kernels for non-parametric two-sample tests. In ICML, 2020.
>
> **We hope the clarifications provided have addressed your concerns. Please feel free to reach out if you have any further questions or require additional details.**

---

### Official Review · Reviewer_GJmC · 2025-07-02

**Clarity:** 3
**Significance:** 2
**Originality:** 3
**Rating:** 5
**Confidence:** 4

**Summary:**

This paper addresses the challenge of optimizing kernel aggregation for nonparametric hypothesis testing.  A new method is introduced named DUAL (U-statistic aggregation with learned kernels). They propose using covariance-informed diversity measures to balance kernel diversity and kernel effectiveness. Theoretical results demonstrate the consistency of the test power and control of the Type I error. Experimental results show the superior performance of the proposed method.

**Questions:**

1.  The paper states that computational cost scales cubically with the number of kernels. This makes using a very large-scale kernel candidate set infeasible. Therefore, when employing a limited feasible set of kernels, how the initialization of this kernel set is performed might significantly impact the final results. Has this initialization sensitivity been explored?
2.  Have the authors considered moving beyond a discrete set of kernel candidates? For instance, could a continuous optimization approach, directly learning the kernel parameters, be a viable alternative to mitigate the computational limitations?
3.  Could the proposed methodology be extended to other kernel-based application domains？

**Ethical Concerns:**

["NO or VERY MINOR ethics concerns only"]

**Final Justification:**

The authors have solved my concerns on kernel initialization and computation complexity.  Thus I consider to raise my score.

**Limitations:**

yes

**Paper Formatting Concerns:**

none.

**Quality:**

3

**Strengths And Weaknesses:**

The research appears comprehensive, with sound methodology and potential practical utility.

A potential limitation, however, is that the approach may not be scalable to scenarios involving very large-scale kernel candidate sets.

---

> ### Author Rebuttal · Authors · 2025-07-31
>
> Thank you very much for your thoughtful and constructive review. We truly appreciate your time and effort. Below, we answer your questions in detail.
>
> **Weakness 1:**
> > A potential limitation, however, is that the approach may not be scalable to scenarios involving very large-scale kernel candidate sets.
>
> **Response 1:**
>
> Thank you for the question regarding the scalability of our method. It is true that using a very large candidate kernel pool may lead to computational inefficiency. However, in the context of two-sample and independence testing, increasing the number of kernels beyond a moderate size does not lead to noticeable gains in test power. As shown in the empirical results from [1], specifically Figure 6 in Section 5.7, increasing the number of kernels from 10 to 100 and even to 1000 does not improve power. **This supports the choice of using a small number of kernels (e.g., $c = 10$)**, as there is nothing to gain (empirically) from using a finer discretization for the bandwidths of kernels. In fact, the referenced study even considers aggregating 12,000 kernels. We will add a discussion in the next version on this point to illustrate the scalability of our method.
>
> **Question 1:**
> > The paper states that computational cost scales cubically with the number of kernels. This makes using a very large-scale kernel candidate set infeasible. Therefore, when employing a limited feasible set of kernels, how the initialization of this kernel set is performed might significantly impact the final results. Has this initialization sensitivity been explored?
>
> **Response 1:**
>
> Thanks for the question on the cubical computational cost and the initialization of kernel sets. We would like to correct the time complexity reported in the original manuscript. Specifically, the complexity should be $O((n^2c^2 + c^3)B)$, rather than $O(Bc^3n^2)$, where $n$ is the sample size, $c$ is the number of kernels, and $B$ is the number of wild bootstrap iterations. **Notably, the $c^3$ term represents the complexity of matrix solves involving the estimated covariance matrix $\hat{\Sigma}_{H_0}$, which is independent of the sample size $n$**. As previously discussed, using a small number of kernels (e.g., $c = 10$) has been shown to be sufficient in practice [1].
>
> **For initializing the kernel set, we follow the methodology of [1,2,3] and adopt a heuristic-based approach.** Specifically, we compute the 0.05 and 0.95 quantiles of the pairwise distances between samples, and then select bandwidths uniformly between the minimum of the 0.05 quantile and the maximum of the 0.95 quantile. With a fixed kernel set size $c$, this procedure yields the same kernel set for the same data, ensuring both performance and reproducibility.
>
> In the following table, we further analyze the sensitivity of the kernel set initialization. Specifically, we compare the test powers of the MMD-DUAL, MMDAgg, and MMD-FUSE methods when the kernel sets (with 10 kernels) are initialized using either the **heuristic approach or random initialization:**
> | Test power | BLOB | | | | | |
> |------------|------|-------|-------|-------|-------|-------|
> |**Heuristic**| N=50 | N=100 | N=150 | N=200 | N=250 | N=300 |
> | MMD-DUAL | 0.134±0.016 | **0.454±0.022** | **0.750±0.028** | **0.915±0.016** | **0.983±0.005** | **0.998±0.001** |
> | MMDAgg | 0.124±0.055 | 0.270±0.076 | 0.539±0.073 | 0.764±0.053 | 0.941±0.026 | 0.987±0.008 |
> | MMD-FUSE | **0.153±0.055** | 0.346±0.065 | 0.673±0.084 | 0.863±0.042 | 0.972±0.011  | **0.998±0.002** |
> |**Random**|  |  |  |  |  |  |
> | MMD-DUAL | 0.084±0.067 | **0.365±0.123** | **0.420±0.084** | **0.585±0.038** | **0.737±0.055** | **0.825±0.061** |
> | MMDAgg | 0.058±0.028 | 0.148±0.054 | 0.210±0.056 | 0.410±0.163 | 0.675±0.158 | 0.733±0.169 |
> | MMD-FUSE | **0.095±0.067** | 0.098±0.048 | 0.203±0.152 | 0.439±0.181 | 0.656±0.163 | 0.740±0.083 |
>
>
> From the table, we observe the following:
> - **Heuristic initialization:** Across all methods, heuristic initialization yields higher and more stable test power, as indicated by lower standard deviations, compared to random initialization.
> - **Random initialization:** Randomly initializing the kernel sets leads to a noticeable drop in performance for all methods, generally resulting in **lower test power and higher standard deviations**. However, MMD-DUAL differs from MMDAgg and MMD-FUSE in that it performs kernel optimization. **While MMDAgg and MMD-FUSE fully rely on the initial kernel sets, MMD-DUAL can adaptively optimize over a diverse kernel pool even when the initialization is not ideal**. This enables MMD-DUAL to maintain relatively high and stable test power under random initialization.
>
> **We will emphasize that we follow the methodology of [1,2,3] in adopting a heuristic-based initialization approach, and we will include these experimental details and dicussions in the updated version of the paper.**
>
> **Question 2:**
> > Have the authors considered moving beyond a discrete set of kernel candidates? For instance, could a continuous optimization approach, directly learning the kernel parameters, be a viable alternative to mitigate the computational limitations?
>
> **Response 2:**
>
> Thanks for the valuable insights about continuous kernel parameter optimization. As we apply four different categories of kernels (Gaussian, Laplacian, Mahalanobis, and Deep), unifying these into a single parameterized kernel family and learning the parameters jointly is indeed a promising direction to improve computational efficiency. We will discuss this exciting future direction in our next version, focusing on how to simultaneously unify these different kernels while maintaining representational diversity.
>
> **Question 3:**
> > Could the proposed methodology be extended to other kernel-based application domains？
>
> **Response 3:**
>
> Thanks for this open question about the application of our proposed method. Since MMD-DUAL and HSIC-DUAL are general two-sample and independence testing methods that do not rely on any strict assumptions, they can both be employed in real-world kernel-based applications.
>
> For example, MMD-DUAL can be effectively applied to machine-generated text detection [7], where detecting subtle distributional differences between human-written and LLM-generated texts is critical. Similarly, HSIC-DUAL is applicable in machine unlearning evaluation [8], where independence testing can be used to verify whether a model has forgotten specific training data.
>
> Moreover, we stress that our newly proposed statistics with learned diverse kernels can also be used in more general settings beyond hypothesis testing. They can improve any method based on MMD or HSIC, for which the kernel selection problem is crucial. These discrepancies are often used as training objectives in Machine Learning.
>
> As statistical testing-based approaches become increasingly popular in emerging domains due to their theoretical guarantees, they have strong potential for extension to broader applications. We will add this discussion in our next version to illustrate the broader utility of our method beyond conventional testing scenarios.
>
> ---
> [1]. A. Schrab, I. Kim, M. Albert, B. Laurent, B. Guedj, and A. Gretton. MMD aggregated two-sample test. Journal of Machine Learning Research, 2023.
>
> [2]. F. Biggs, A. Schrab, and A. Gretton. MMD-Fuse: Learning and combining kernels for two sample testing without data splitting. In NeurIPS, 2023.
>
> [3]. A. Schrab, I. Kim, B. Guedj, and A. Gretton. Efficient aggregated kernel tests using incomplete U-statistics. In NeurIPS 2022.
>
> [4]. A. Gretton, K. M. Borgwardt, M. J. Rasch, B. Schölkopf, and A. J. Smola. A kernel method for the two-sample-problem. In NeurIPS, 2006.
>
> [5]. A. Gretton, K. Fukumizu, C. H. Teo, L. Song, B. Schölkopf, and A. J. Smola. A kernel statistical test of independence. In NeurIPS, 2007.
>
> [6]. F. Liu, W. Xu, J. Lu, G. Zhang, A.Gretton, and D. J. Sutherland. Learning deep kernels for non-parametric two-sample tests. In ICML, 2020.
>
> [7]. S. Zhang, Y. Song, J. Yang, Y. Li, B. Han, and M. Tan. Detecting machine-generated texts by multi-population aware optimization for maximum mean discrepancy. In ICLR, 2024.
>
> [8]. R. Mehta, S. Pal, V. Singh, S. N. Ravi. Deep Unlearning via Randomized Conditionally Independent Hessians. In CVPR, 2022.
>
> **We hope our response has clarified the confusion and addressed your concerns. We would greatly appreciate it if you could kindly reconsider your score. Thank you.**

---

> > ### Author Response · Authors · 2025-08-06
> > **Appreciation and willingness to clarify any remaining questions!**
> >
> > Dear reviewer GJmC,
> >
> > Thank you again for taking the time to review our submission. We would like to kindly check if you have any further questions or concerns that remain unaddressed following our response. We would be more than happy to provide additional clarifications or engage in further discussion if needed.
> >
> > We truly appreciate your insights, and we sincerely hope that our clarifications have helped address any remaining concerns and contribute positively to your overall assessment.
> >
> > Best regards,
> > Authors of Submission 16150

---

> > ### Comment · Reviewer_GJmC · 2025-08-07
> >
> > Thank you for your comprehensive response to my review comments. I apologize for the delay in my follow-up.
> > While you've addressed most of my concerns, I would encourage you to expand the discussion to include broader perspectives on flexible kernel approaches beyond multiple kernels.
> > I will raise my overall score.

---

> ### Author Response · Authors · 2025-08-07
> **Thanks for raising the overall score!**
>
> Thank you very much for your thoughtful review and for raising your overall score. We greatly appreciate your valuable feedback and the time you've invested in evaluating our work. We will add the discussion of how our proposed method can be extended to other kernel-based application field in the updated version. Your insights will help strengthen the contribution and its positioning within the broader literature.
>
> Authors of Submission 16150

---

### Official Review · Reviewer_67oQ · 2025-07-02

**Clarity:** 3
**Significance:** 3
**Originality:** 3
**Rating:** 5
**Confidence:** 4

**Summary:**

The manuscript proposed a kernel aggregating methods in hypothesis testing aiming at improves kernel-based nonparametric tests by utilizing the diversity of kernels inspired by ensemble learning,

**Questions:**

Can the author provide a guarantee for the error rate of the test based on the limited sample size? If so, it will significantly enhance readers' understanding of the usability of their method.

**Ethical Concerns:**

["NO or VERY MINOR ethics concerns only"]

**Final Justification:**

I maintain my previous assessment.

**Quality:**

3

**Strengths And Weaknesses:**

Strengths: The motivation behind the method is natural, and its effectiveness is guaranteed by certain theoretical analyses.

Weaknesses: Its calculation has a relatively high level of complexity (O(B*c^3*n^2)). In addition, its theoretical guarantee is based on an infinite sample. In reality, people are more concerned about how the error rate of the test is controlled when using a limited n.

---

> ### Author Rebuttal · Authors · 2025-07-31
>
> Thank you very much for your thoughtful and constructive review. We truly appreciate your time and effort. Below, we answer your questions in detail.
>
> **Weakness 1:**
> > Its calculation has a relatively high level of complexity $(O(Bc^3n^2))$.
>
> **Response 1:**
>
> Thank you for noticing the time complexity. Here, we first revise and correct the expression of the time complexity as follows:
> $$
> O((n^2(c^2+c)+c^3)B)=O((n^2c^2+c^3)B),
> $$
> where $n$ denotes the sample size, $c$ is the number of kernels and $B$ is the number of wild bootstrap iterations. In this expression: $n^2c^2$ corresponds to the complexity of computing the elements of the $c \times c$ covariance matrix; $n^2c$ accounts for the complexity of evaluating MMD statistics using $c$ kernels; $c^3$ represents the complexity of matrix solves involving the estimated covariance matrix $\hat{\Sigma}_{H_0}$.
>
> Second, the time complexity is cubic in terms of the number of selected kernels, i.e., $c$. However, in the context of two-sample and independence testing, increasing the number of kernels, i.e., $c$, beyond a moderate size does not lead to noticeable gains in test power. As shown in the empirical results from [1], specifically Figure 6 in Section 5.7, increasing the number of kernels from 10 to 100 and even to 1000 does not improve power. In fact, the referenced study even considers aggregating 12,000 kernels. This supports the choice of using a small number of kernels (e.g., $c = 10$), as there is nothing to gain empirically from using a finer discretization of kernel bandwidths. With such a small number of kernels, the time complexity is rarely a bottleneck in practice.
>
> We will add a discussion in the updated version on this point to illustrate the scalability of our method.
>
> **Weakness 2 + Question 1:**
> > In addition, its theoretical guarantee is based on an infinite sample. In reality, people are more concerned about how the error rate of the test is controlled when using a limited $n$.
>
> > Can the author provide a guarantee for the error rate of the test based on the limited sample size? If so, it will significantly enhance readers' understanding of the usability of their method.
>
> **Response 2 + 1:**
>
> Thanks for raising this important question. In Theorem 1, we establish exact type-I error control for any sample size, indicating that the type-I error is maintained at the level $\alpha$ even with the sample size $n$ is limited.
>
> We note that classical MMD/HSIC tests benefit from finite sample power guarantees taking the form of uniform separation rates with Sobolev regularity assumption [1,2]. However, to the best of our knowledge, it remains open to theoretically analyze the exact sample complexity of the type‑II error (i.e., 1 minus the test power) when accounting for the covariances between statistics computed with different kernels. Although we have endeavored to address this, the primary challenge lies in the technical characterization of those covariance dependencies. We view this as a promising direction for future work and will add a discussion on this point in the next version.
>
> ---
>
> [1]. A. Schrab, I. Kim, M. Albert, B. Laurent, B. Guedj, and A. Gretton. MMD aggregated two-sample test. Journal of Machine Learning Research, 2023.
>
> [2]. M. Albert, B. Laurent, A. Marrel, et al. Adaptive test of independence based on HSIC measures. The Annals of Statistics, 2022, 50(2): 858-879.
>
> **We hope the clarifications provided have addressed your concerns. Please feel free to reach out if you have any further questions or require additional details.**

---

> > ### Comment · Reviewer_67oQ · 2025-08-07
> > **Official Comment by Reviewer 67oQ**
> >
> > Thank you for your clarifications.

---

> ### Author Response · Authors · 2025-08-07
>
> Thank you for confirming that our clarifications have addressed your concerns.
>
> Authors of Submission 16150

---

### Official Review · Reviewer_fLJY · 2025-07-03

**Clarity:** 3
**Significance:** 3
**Originality:** 3
**Rating:** 5
**Confidence:** 2

**Summary:**

This paper aims to improve kernel-based two-sample and independence tests. Existing methods consider aggregating multiple kernels to increase statistical power, but often suffer from selecting non-diverse kernels, thereby limiting its effectiveness. To overcome this, the authors propose a new method, called DUAL, which is a novel test statistic that explicitly promotes kernel diversity by considering the covariance between different kernels. Furthermore, it introduces a selection inference framework that uses information from a training set to select a subset of kernels that are not only diverse but also individually powerful. Empirically, the proposed method is shown to achieve better performance on kernel two-sample tests on 3 datasets and kernel independence tests on 3 datasets, respectively.

**Questions:**

### Q1: run-time overhead ###
While the proposed method DUAL can improve the test power, it also needs to pay the price of run-time ovehead
- Q1.1: What's the time complexity of DUAL for the training stage (Sec 5.1)?
- Q1.2: What's the time complexity of DUAL for the testing stage (Sec 5.2)?
- Q1.3: How does those compared to the baselines, such as MMDAgg and MMD-FUSE?

It would be great if the authors can organize these comparison in the Table.

### Q2: Sample size in Figure 3 ###
- Q2.1: Any potential explanation why the sample size N in Fig-3(b) is so small (<100), compared to Fig-3(c) which is ~1k?
- Q2.2: What's the feature dimensionality for MNIST and CIFAR10? Do you use raw image, or some learned embedding?

### Q3: What are the trainable parameters in Section 5.1? ###
- Q3.1: What's the learnable/trainable parameters for kernel learning/selection in Eq(5) of Section 5.1?
- Q3.2: Its more like an open-ended question, just for the thought process. Can we learn various encoders such that the learned features are somehow orthogonal, which may help the multiple kernel aggregations?

### Q4: Can DUAL help learn better generative models? ###
This is another open research question. I am curious if DUAL, compared to say the simple MMD loss, can help learn better generative models?

**Ethical Concerns:**

["NO or VERY MINOR ethics concerns only"]

**Final Justification:**

Most of my concerns and questions are addressed during the rebuttal. I remained my positive rating for this submission.

**Limitations:**

No potential negative societal impact

**Paper Formatting Concerns:**

No major formatting issue.

**Quality:**

3

**Strengths And Weaknesses:**

### Strengths ###
- S1: The paper writing is well-organized and pleasant to read
- S2: The proposed method, DUAL, is well-motivated, novel, and technical sound

### Weaknesses ###
- W1: DUAL introduces run-time overhead for both the training and inference stage

---

> ### Author Rebuttal · Authors · 2025-07-31
>
> Thank you very much for your thoughtful and constructive review. We truly appreciate your time and effort. Below, we answer your questions in detail.
>
> **Weakness 1 + Question 1:**
> > DUAL introduces run-time overhead for both the training and inference stage
>
> > While the proposed method DUAL can improve the test power, it also needs to perform extra computation.
> >- Q1.1: What’s the time complexity of DUAL for the training stage (Sec 5.1)?
> >- Q1.2: What’s the time complexity of DUAL for the testing stage (Sec 5.2)?
> >- Q1.3: How do these compare to the baselines, such as MMDagg and MMD‑FUSE?
> >
> > It would be great if the authors could organize these comparisons in a table.
>
> **Response 1 + 1:**
>
> Thanks for pointing out this question. We thoroughly discuss the complexity and compare it with previous methods as follows.
>
> *Training Phase (Using the Adam Optimizer)*
>
> **MMD-DUAL:** $O((n^2(c^2+c)+c^3+T)*M)=O((n^2c^2+c^3+T)*M)$, where $n$ denotes the sample size, $c$ is the number of kernels, $T$ is the number of kernel parameters, and $M$ represents the number of optimization epochs. In this expression: $n^2c^2$ corresponds to the complexity of computing the elements of the $c \times c$ covariance matrix; $n^2c$ accounts for the complexity of evaluating MMD statistics using $c$ kernels; $c^3$ represents the complexity of matrix solves involving the estimated covariance matrix $\hat{\Sigma}_{H_0}$.
>
> **MMDAgg:** N/A (no training phase)
>
> **MMD-FUSE:** N/A (no training phase)
>
> *Testing Phase*
>
> **MMD-DUAL:** $O((n^2c^2+c^3)B)$, where $n^2c^2 + c^3$ corresponds to the complexity of computing the aggregated statistics, and $B$ is the number of wild bootstrap iterations.
>
> **MMDAgg:** $O(n^2c(B+B'))$, where $n^2c$ corresponds to the time complexity of calculating MMD statistics using $c$ kernels, $(B+B')$ is the total number of wild bootstrap iterations and permutations, typically with $B=B'$.
>
> **MMD-FUSE:** $O(n^2cB)$, where $n^2c$ corresponds to the time complexity of calculating MMD statistics using $c$ kernels, $B$ is the number of wildbootstrap iterations.
>
> | ***Time Complexity*** | MMD-DUAL| MMDAgg| MMD-FUSE |
> |--|--|--|--|
> |Training| $O((n^2c^2+c^3+T)*M)$ | N/A | N/A |
> |Testing | $O((n^2c^2+c^3)B)$ | $O(n^2c(B+B'))$ | $O(n^2cB)$ |
>
> The time complexity primarily depends on the sample size factor $n^2$. Other parameters, such as the kernel size $c$, the number of optimization parameters $T$, the number of optimization epochs $M$, and the number of wild bootstrap iterations $B$, are considered fixed and do not vary with the sample size.
>
> We further present the running times of MMD-DUAL, MMDAgg, and MMD-FUSE across different datasets and sample sizes, as shown in the table below. In this experiment, all three methods are initialized with the same kernel set, following the experimental setup of [4]. Moreover, since MMD-DUAL involves a training process, we follow the procedure in [1] and perform a half-half train-test split. Compared to the MMDAgg and MMD-FUSE which directly testing on the whole samples, MMD-DUAL performs the testing prcoess only on half of the data.
>
> | Running Times (s) | BLOB | | | | | | MNIST | | | | | | ImageNet | | | | | |
> |--|--|--|---|----|----|---|---|-|---|---|---|---|-----|----|----|---|---|---|
> | **Training** | N=50 | N=100 | N=150 | N=200 | N=250 | N=300 | N=20 | N=30 | N=40 | N=50 | N=60 | N=70 | N=400 | N=500 | N=600 | N=700 | N=800 | N=900 |
> | MMD-DUAL | 0.04692 | 0.04774 | 0.04834 | 0.04989 | 0.05183 | 0.05418 | 1.29088 | 1.32365 | 1.34037 | 1.35724 | 1.36015 | 1.37741 | 9.68179 | 11.08146 | 13.54574 | 15.99571 | 18.72705 | 21.98263  |
> | MMDAgg | 0 | 0 | 0 | 0 | 0 | 0 | 0 | 0 | 0 | 0 | 0 | 0 | 0 | 0 | 0 | 0 | 0 | 0 |
> | MMD-FUSE | 0 | 0 | 0 | 0 | 0 | 0 | 0 | 0 | 0 | 0 | 0 | 0 | 0 | 0 | 0 | 0 | 0 | 0 |
> | **Testing** | | | | | | | | |  | | | | | | | | | |
> | MMD-DUAL | 0.00395 | 0.00403 | 0.00406 | 0.00417 | 0.00421 | 0.00429 | 0.01778 | 0.01782 | 0.01791 | 0.01818 | 0.01821 | 0.01837 | 0.70675 | 0.80746 | 1.02429 | 1.10982 | 1.27423 | 1.47108 |
> | MMDAgg | 0.00355 | 0.00372 | 0.00382 | 0.00402 | 0.00427 | 0.00431 | 0.02936 | 0.03051 | 0.03079 | 0.03103 | 0.03111 | 0.03163 | 0.76292 | 0.95655 | 1.16676 | 1.33279 | 1.44765 | 1.59419 |
> | MMD-FUSE | 0.00141 | 0.00174 | 0.00188 | 0.00246 | 0.00294 | 0.00389 | 0.02768 | 0.02792 | 0.02817 | 0.02823 | 0.02827 | 0.02831 | 0.67961 | 0.84282 | 0.98464 | 1.17171 | 1.40641 | 1.55239 |
>
> From the table, we observe that all methods have similar testing times. The key difference lies in the training phase: MMDAgg and MMD-FUSE do not involve any optimization and therefore incur no training time, while MMD-DUAL includes a kernel optimization phase, resulting in a training cost.We will add the above time complexity analysis and empirical running time comparisons in our updated version.
>
>
> **Question 2:**
> > Sample Size in Figure 3
> >- Q2.1: Any potential explanation why the sample size _N_ in Fig. 3(b) is so small (< 100), compared to Fig. 3(c) which is ∼ 1 k?
> >- Q2.2: What’s the feature dimensionality for MNIST and CIFAR‑10? Do you use raw images or some learned embedding?
>
> **Response 2:**
>
> Thanks for your questions about the sample size and feature dimensionality in our experimental results.
>
> Regarding the sample size, the reason why $N$ in Fig. 3(b) and Fig. 3(c) differ is due to the dataset complexity. As discussed in [1], compared to the more challenging dataset ImageNet, two-sample testing methods generally perform better on the simpler dataset MNIST vs. MNIST.Fake [2]. This is empirically supported in [3], where achieving a converging test power of 1 on MNIST requires significantly fewer samples than on ImageNet.
>
> For input feature dimensionality, we resize MNIST images to $32\times32\times1$ and ImageNet images to $32\times32\times3$, following the standard experimental settings used in previous two-sample testing works [1,3,4]. We use raw images as input. When the deep kernel from [1] is leveraged, a neural network is trained from scratch to learn feature embeddings, transforming raw inputs into learned representations. We will include these implementation details in the next version to enhance reproducibility.
>
> **Question 3:**
>
> > Trainable Parameters in Section 5.1
> >- Q3.1: What are the learnable/trainable parameters for kernel learning/selection in Eq.(5) of Section 5.1?
> >- Q3.2: (Open‑ended) Can we learn various encoders such that the learned features are orthogonal, which may help the multiple‑kernel aggregations?
>
> **Response 3:**
>
> Thanks for the questions about the learnable parameters for kernel learning. As we use four different kinds of kernels, we will list the parameters for each kernel below:
> - Gaussian kernel: $κ(x, y) = \exp(-||x-y||^2_2/σ^2)$, where bandwidth $σ$ is the trainable parameter.
> - Laplacian kernel: $κ(x, y) = \exp(-||x-y||_1/σ)$, where bandwidth $σ$ is the trainable parameter.
> - Mahalanobis kernel: $κ(x, y) = \exp(-(x-y)^{T}Σ^{-1}(x-y))$, where the covariance matrix $Σ$ is the trainable parameter.
> - Deep kernel: $κ(x, y) = [(1-ϵ)\exp(-||ϕ_{ω}(x) - \phi_{ω}(y)||^2_2/σ_{ϕ}^2) +ϵ]\exp(-||x-y||^2_2/σ^2)$, where the parameter $ω$ in the deep neural network $ϕ_{ω}$, bandwidths $σ_{ϕ}$ and $σ$, and the weight $ϵ$ are all trainable.
>
> We appreciate the suggestion and will incorporate these details in the supplementary materials to improve clarity and reproducibility, in addition to the references already cited [1,5,6].
>
> For the open questions, it is an interesting idea that orthogonal learned representations from multiple encoders contain the most diverse information from each other. This technique can indeed be compatible with and applied to our method to enhance the test power of multi-kernel aggregation, but we should be cautious about the trade-off between the training samples used for learning the encoders and the number of samples reserved for testing.
>
>
>
> **Question 4:**
> > Can DUAL Help Learn Better Generative Models?
> This is another open research question. I am curious if DUAL, compared to the simple MMD loss, can help learn better generative models?
>
> **Response 4:**
>
> As noted in prior work, MMD-GAN [7] is a generative model that relies on the standard MMD loss. In principle, we can replace this loss with our proposed multi-variate $U$-statistics in Eq.(4), i.e., $T_n^K(W)$. This statistic aggregates information from MMD statistics computed with multiple kernels, enabling it to capture richer information about the difference between the target and generative distributions. By minimizing $T_n^K(W)$, the generator potentially produces samples that better match the target distribution compared to using the standard MMD alone. This suggests a promising direction for improving generative models, and we will include this discussion as part of future work in the next version.
>
> ---
>
> [1]. F. Liu, W. Xu, J. Lu, G. Zhang, A. Gretton, and D. J. Sutherland. Learning deep kernels for non-parametric two-sample tests. In ICML, 2020.
>
> [2]. A. Radford, L. Metz, and S. Chintala. Unsupervised representation learning with deep convolutional generative adversarial networks. In ICML, 2016.
>
> [3]. X. Tian, L. Peng, Z. Zhou, M. Gong, A. Gretton, F. Liu. A unified data representation learning for non-parametric two-sample testing. In UAI, 2025.
>
> [4]. A. Schrab, I. Kim, M. Albert, B. Laurent, B. Guedj, and A. Gretton. MMD aggregated two-sample test. Journal of Machine Learning Research, 2023.
>
> [5]. A. Chatterjee and B. B. Bhattacharya. Boosting the power of kernel two-sample tests. Biometrika, 2025.
>
> [6]. Z. Zhou, J. Ni, J. Yao, and W. Gao. On the exploration of local significant differences for two-sample test. In NeurIPS, 2023.
>
> [7]. C. Li, W. Chang, Y. Cheng, Y. Yang, B. Póczos. MMD GAN: Towards Deeper Understanding of Moment Matching Network. In NeurIPS, 2017.
>
> **We hope the clarifications provided have addressed your concerns. Please feel free to reach out if you have any further questions or require additional details.**

---

### Decision · Program_Chairs · 2025-09-17

**Decision:**

Accept (poster)

**Comment:**

This paper presents a novel framework, DUAL (Diverse U-statistic Aggregation with Learned Kernels), to enhance kernel-based nonparametric hypothesis testing, specifically targeting two-sample and independence tests. The core innovation lies in promoting kernel diversity alongside individual kernel effectiveness. The authors provide solid theoretical support, including Type-I error control and power consistency.

Strengths：All reviewers agree that the paper is technically sound, well-structured, and supported by rigorous theoretical analysis and comprehensive experimental validation. The writing is clear and pleasant to read, making the methodology and contributions easy to follow. Reviewer BHAJ particularly appreciates the combination of theoretical rigor and clear empirical improvements over existing kernel aggregation methods.

Weaknesses：The reviewers primarily raised concerns regarding the computational overhead and scalability of the proposed method, especially when the number of kernels c becomes large. In addition, Reviewer BHAJ expressed concerns about the sensitivity of the method to initialization or the number of kernels.

The most important reasons for the decision：This paper addresses a practical limitation in kernel aggregation methods. The proposed methods are both theoretically solid and experimentally well grounded.

Summarization of the discussion and changes during the rebuttal period:  Some reviewers raised concerns about the scalability of the method, which the authors addressed with additional experiments and supportive literature in the rebuttal.

Overall comments:

After the rebuttal phase, all reviewers raised their score to accept, resulting in unanimous acceptance. The paper investigates aggregated kernel-based nonparametric hypothesis testing. All reviewers agree that the proposed methods are both theoretically solid and experimentally well grounded. On the other hand, concerns remain regarding the computational overhead and scalability of the proposed method, especially when the number of kernels c becomes large. Overall, the quality of the work is high and solid, and it is likely to be of interest to the community. Therefore, I recommend acceptance as a poster.